# The interplay of habitat quality and temperature shape demographic patterns of mule deer (*Odocoileus hemionus*) in North America

William M. Janousek [1]✉, Aaron N. Johnston [2], Sarah L. Bullock[3], Sarah R. Dewey[4], L. Embere Hall [5], Teagan A. Hayes[1,10], Katey S. Huggler [6,7,9], Matthew J. Kauffman [8], Tayler N. LaSharr[6], Blake Lowrey[2], Rhiannon P. Jakopak[6], Kevin L. Monteith[6] & Tabitha A. Graves [1]

Mule deer (*Odocoileus hemionus*) are declining in abundance across their broad distribution in western North America. Identifying drivers of mule deer demography could inform habitat restoration. However, linking habitat quality to vital rates is challenging and often done indirectly using proxy metrics. We combine habitat selection with climate-related effects to identify synergistic influences affecting mule deer age ratios (fawn:doe). We used location data from 1473 female deer over 22 years in Wyoming to fit seasonal resource selection models, predict habitat suitability, and model age ratios as a function of drought conditions, winter severity, and seasonal habitat. Here we show temperature had the largest effect on mule deer recruitment with age ratios declining following hotter summers and colder winters. Age ratios increased with higher proportions of habitat with high-quality summer habitat of particular importance. Given the likely increases in summer temperatures and extreme winter weather events, populations may struggle to increase recruitment over the next half-century. Targeted management supporting forage quantity and quality, especially on summer range, could buffer the effects of decades-long drought conditions. Our findings also indicate mule deer avoid areas with high densities of oil and gas development. By delineating important mule deer habitat, we offer spatial tools for development siting and mitigation in Wyoming and a framework for broader application across the western United States.

The interactions of forage resources, seasonal weather, long-term climatic variation and predation risk all serve to influence ungulate population demography by driving key vital rates such as the probability of pregnancy[1,2] and offspring and adult survival[3,4]. Changes in habitat have been linked to declines in ungulate populations through multiple (often interacting) mechanisms, including the loss of migration routes between seasonal ranges[5], trade-offs between forage resources and predation risk[6], mismatches between parturition dates and greening vegetation[7], aridification[8], and overall habitat fragmentation[9]. Given the strong influence of abiotic conditions and habitat quality on demography, degradation in conditions poses significant challenges to supporting populations of many ungulate species.

Mule deer (*Odocoileus hemionus*), an ungulate species native to western North America, are of conservation concern given their overall

---

[1]U.S. Geological Survey, Northern Rocky Mountain Science Center, West Glacier, MT, USA. [2]U.S. Geological Survey, Northern Rocky Mountain Science Center, Bozeman, MT, USA. [3]Bureau of Land Management, Renewable Resources Branch, Cheyenne, WY, USA. [4]National Park Service, Grand Teton National Park, Moose, WY, USA. [5]Wyoming Game and Fish Department, Laramie, WY, USA. [6]Haub School of Environment and Natural Resources, Department of Zoology and Physiology, University of Wyoming, Laramie, WY, USA. [7]Wyoming Cooperative Fish and Wildlife Research Unit, Department of Zoology and Physiology, University of Wyoming, Laramie, WY, USA. [8]U.S. Geological Survey, Wyoming Cooperative Fish and Wildlife Research Unit, Department of Zoology and Physiology, University of Wyoming, Laramie, WY, USA. [9]Present address: Montana Cooperative Wildlife Research Unit, University of Montana, Missoula, MT, USA. [10]Present address: Intermountain West Joint Venture, Missoula, USA. ✉e-mail: wjanousek@usgs.gov

declining populations[10]. Declines in mule deer populations can have both ecological and economic consequences. For example, reduced browsing pressure may alter vegetation communities, leading to shifts in plant composition and structure[11]. These declines can also affect the economy, including the millions of dollars spent annually on big game hunting[12]. A rich history of research (nearly 4000 published studies on mule deer since 1904[13]), investigates the landscape characteristics that shape mule deer habitat across scales and seasons, including vegetation cover[14] and forage resources[15], terrain characteristics[16,17], energy development and human encroachment[18,19], and winter weather[16,20]. Fine-grained resource selection by mule deer indicates navigating these landscapes is a complicated reflection of reproductive state[14,21], time of year[22], and migratory behavior[15]. While this complexity makes it hard to define "suitable habitat" for mule deer, seasonal resource selection function (RSF) modeling has proven to be reliable for understanding habitat selection and informing management[16,19,23].

The mechanisms that underpin vital rates for mule deer can be variable, from nutritional condition to predation pressure[18,24–27]. Juvenile survival during the first year of life may be dependent on maternal body condition during gestation[28,29], habitat selection for greater concealment by the mother during early life[30], and forage resources available post-weaning[28]. Studies often use body condition as a metric to indirectly link habitat quality and vital rates. However, direct links between variation in habitat quality and mule deer demography are not well-documented (but refer to Merems et al.[31]), in part due to the diversity of mechanisms that connect the two and the availability of data at scale to answer such questions. Given the connection between habitat quality, maternal body condition, and fawn survival, variation in habitat quality has the potential to be a dominant driver of offspring recruitment in mule deer and, therefore, a driver of mule deer population dynamics.

We evaluated the relationship between mule deer recruitment and annual variation in habitat suitability during summer and winter for 21 mule deer herds across Wyoming, USA. Our main goal was to quantify the relationship between the amount of seasonal habitat available in the study area and age ratios (i.e., fawn:doe ratios, an index of recruitment of fawns into the fall population) while accounting for other factors with direct and indirect mortality implications such as drought conditions and winter severity which can alter habitat conditions and induce stress[28]. Given that age ratio surveys are primarily conducted in late fall, we sought to assess the relative influence of post-birth summer conditions experienced by fawns compared to the winter conditions their mothers were exposed to during the fawns' gestation. To accomplish this goal, we first had to develop resource selection functions to predict how landscape characteristics define mule deer habitat. Previous work has indicated a strong connection between seasonal weather patterns and offspring recruitment in ungulates[29,32] and deliberated the relative importance of summer vs. winter habitat conditions as the dominant driver of ungulate demography[28,33,34]. After accounting for annual variation in temperature and precipitation, we predicted the proportion of available summer and winter habitat would be positively correlated with fawn:doe ratios. This approach allowed us to compare the importance of different seasonal habitat conditions versus the abiotic drivers of survival and reproduction.

## Results

The global positioning system (GPS) collar data used for RSF modeling included 661486 unique randomly selected daily locations from 1473 female mule deer across 22 years, 2001 to 2022 (Fig. 1). This resulted in a used-available dataset of 1322972 points (60% winter and 40% summer[35]). In total, 3778 unique animal-years were represented with an average of 175 ($\sigma = 90.90$) used locations per animal-year. Migratory deer were represented in a 2:1 ratio compared to non-migratory deer with individual study herds ranging from 3% to 100% migratory (Supplementary Table 1). For the winter RSF analyses the West, Northeast, and Southeast regional subsets comprised 68%, 13%, and 19% of the data, respectively. For summer RSFs

the Northwest, Southwest, Northeast, and Southeast regional subsets comprised 20%, 45%, 16%, and 19%, respectively (Fig. 1).

## Patterns of resource selection

We analyzed mule deer habitat selection in relation to a suite of landscape covariates identified a priori to broadly represent factors such as forage availability, thermoregulation, and predator or human avoidance. Considering covariates with statistically detectable effects, the selection coefficients varied in magnitude across regions but were consistent in direction (Fig. 2, Supplementary Table 2). These results support the idea that mule deer tend to select for similar landscape characteristics across populations, but the relative importance of each unique resource varies spatially.

During summer, selection for non-sagebrush shrub and perennial forb and grass cover coupled with avoidance of annual forb and grass cover aligns with the literature regarding diet of mule deer during this period[36,37]. Deer selected for areas of increased productivity, i.e., higher Normalized Difference Vegetation Index (NDVI) amplitude, and areas that reach peak rate of green-up later in the summer. This supports findings that deer track the progression of forage growth and use summer ranges aligned with peak resource availability[38,39]. We found strong support for avoidance of roads in the more highly populated Western region during summer but no support for this effect in winter in any region; 95% credible intervals (CRIs) cross zero. Mule deer were more likely to use areas with high topographic variability and avoided permanent water bodies which mostly occur on valley floors and have increased rates of human activity. We also found selection for areas with higher heat loading and avoidance of areas with high snow-water equivalents (SWE) in winter. Preliminary models with terrain wetness index failed to fit and this variable was not carried forward into the final RSF formulations. We found negative effects of the density of oil and gas development on selection with slightly stronger avoidance in summer ($\beta = -0.41$, 95% CRI $= -0.58$, $-0.25$) than winter ($\beta = -0.31$, 95% CRI $= -0.38$, $-0.24$), although CRIs overlapped. These results were incorporated as direct additive effects in the seasonal habitat suitability predictions.

Summer suitability had high spatial variability within years (mean spatial standard deviation, 0.20), The most suitable summer habitat tended to occur in more mountainous areas (Fig. 3a), with substantial variability in reaching premium suitability across years in lower elevation areas particularly in the central portion of Wyoming (Fig. 3c). Winter suitability within winter use areas was less spatially variable (mean spatial standard deviation, 0.09, Fig. 3b) with fewer areas that tended to be less suitable and more variable in attaining suitability over time (Fig. 3d). At the statewide level, we found a positive trend in the proportion of premium summer habitat ($\beta = 0.02$, $p < 0.001$, Fig. 4a) resulting in a collective net increase in premium summer habitat area of 42,486 km² (16,404 mi²) from 2000 to 2023. This likely represents the conversion of summer habitat from suitable to premium rather than the creation of new habitat. We did not detect a trend in the amount of suitable winter habitat at the statewide level ($\beta = 0.001$, $P = 0.59$, Fig. 4a).

Most herd units, 88% (n = 30), showed increasing trends in premium summer habitat, and only one herd unit, North Bighorn, exhibited declines in summer habitat (Fig. 4b, Supplementary Table 3). Changes in suitable winter habitat were more variable across herd units with positive trends in two herd units and negative trends in three herd units (Fig. 4c, Supplementary Table 3). The most severe rate of decline in suitable winter habitat was in the North Converse herd unit ($\beta = -0.06$, $P < 0.001$). The modeled declines of winter habitat are likely mainly driven by the severe 2022–2023 winter. For example, in the North Converse unit, the proportion of suitable winter habitat ranged between 0.87 and 0.99 across years except in 2022, when it dropped to 0.71. These results underscore the potential importance of interannual variability in climatic conditions in driving winter habitat suitability in addition to the direct physiological effects of winter severity on mule deer body condition.

**Fig. 1 | Locations of individual mule deer animal-year centroids and corresponding regional assignments.** Locations shown for summer (**a**) and winter (**b**) use areas for female mule deer in Wyoming, USA from 2001 to 2022. The background relief is based on the SRTM 90 m digital elevation model[67].

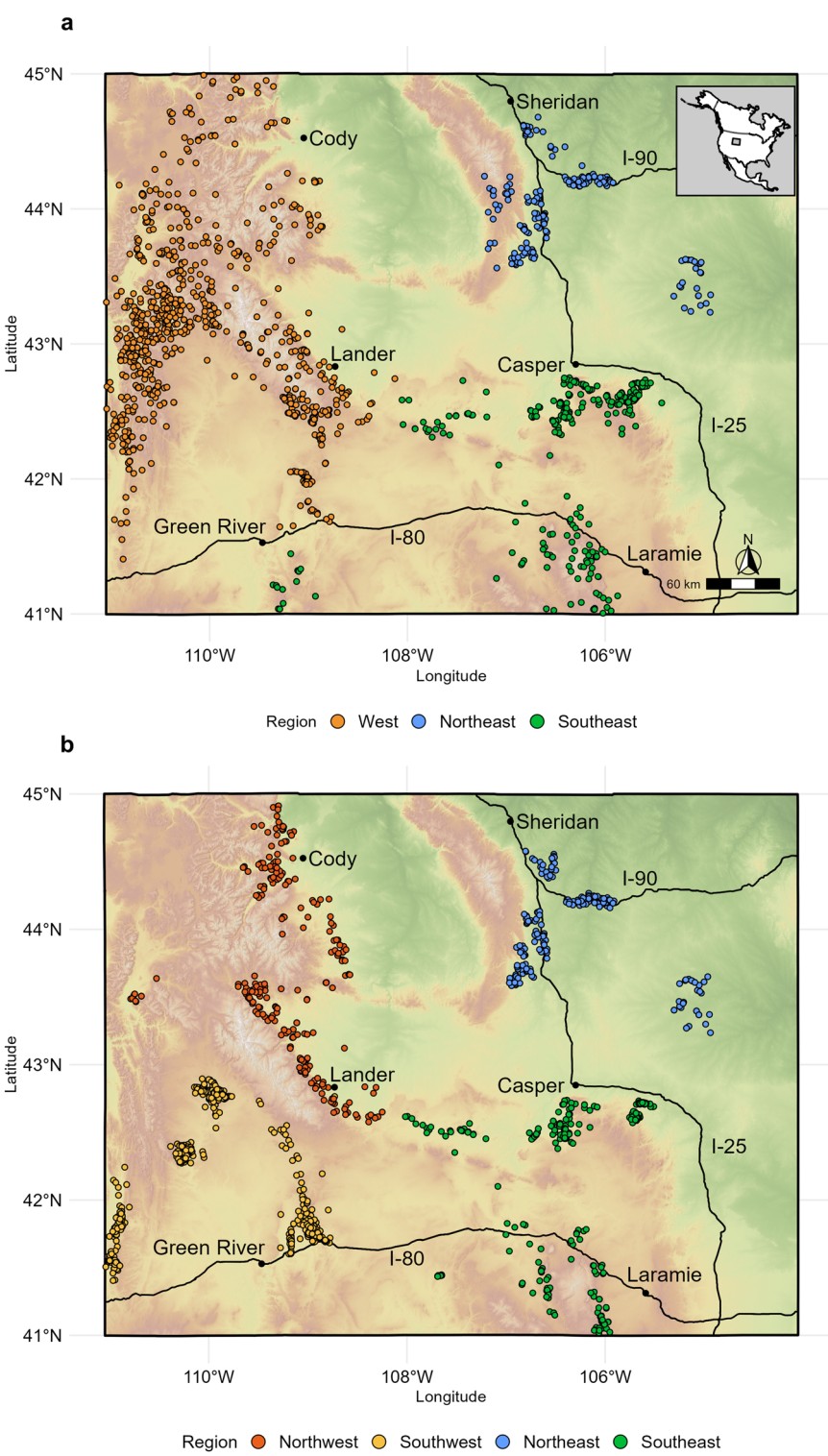

## Environmental influences on fawn:doe ratios

We modeled changes in age ratios as a function of habitat suitability and other abiotic factors from 2001 to 2023 across the state of Wyoming. The top model relating environmental variables to changes in age ratios included the proportion of premium habitat in summer, the proportion of suitable habitat in winter, summer temperature, and winter temperature. We excluded summer precipitation due to its correlation with the proportion of summer habitat ($r = 0.58$). The model with all variables ranked lowest in leave-one-out cross-validation information criterium

(looic, Supplementary Tables 4, 5), and we made inference from these results (Fig. 5). Age ratios increased with warmer winter temperatures and larger proportions of premium summer and suitable winter habitat. Age ratios decreased with higher summer temperature. The effect of summer temperature was the largest overall in absolute magnitude (Fig. 5). The magnitude of the effect size of summer temperature was 1.3 times larger than the proportion of summer habitat while the effect size of winter temperature was 2.5 times larger than the proportion of winter habitat. The effect size of summer habitat was 2.1 times larger than

**Fig. 2 | Resource selection function estimation for third-order selection.** Modeling results separated by regional groups on summer (**a**) and winter (**b**) use areas of female mule deer in Wyoming, USA from 2001 to 2022. Points indicate the median of the posterior distribution, and vertical lines represent the 95% credible interval.

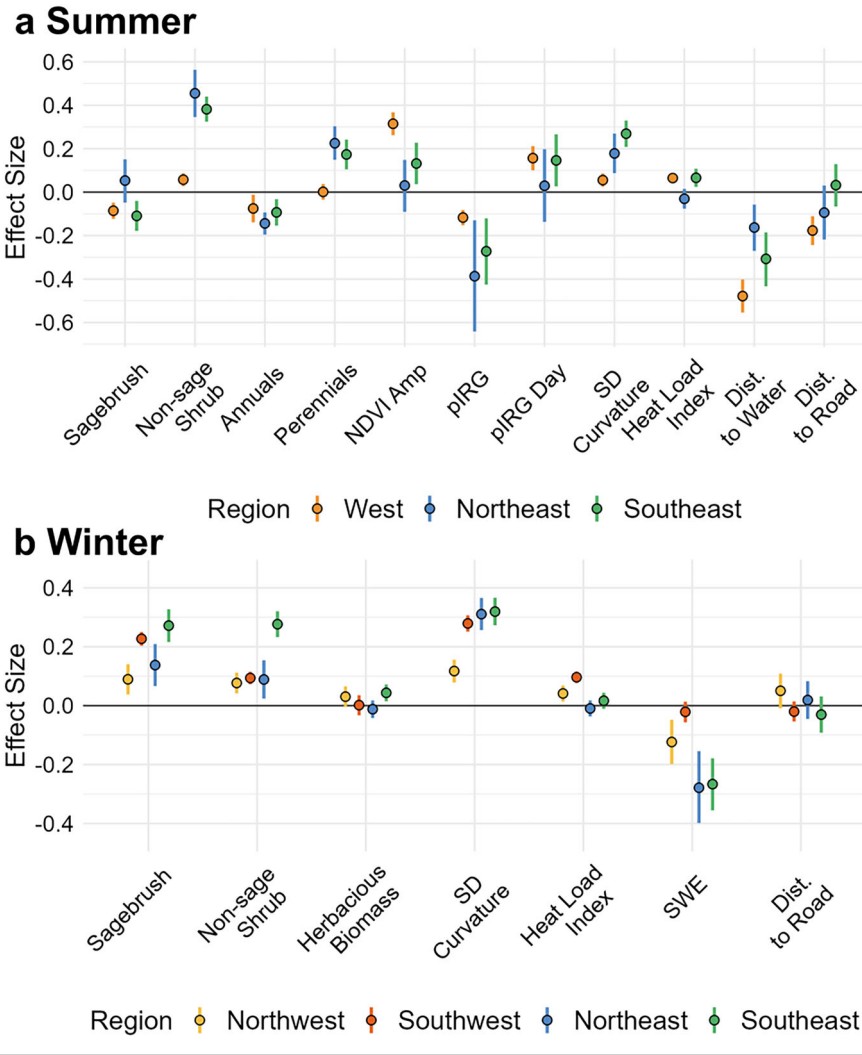

winter habitat while the effect size of summer temperature was 1.1 times larger than the winter temperature.

## Discussion

By combining habitat selection with climate-related effects we identify synergistic abiotic and biotic forces driving mule deer demography in Wyoming, USA. Temperature across seasons proved to be the strongest factors influencing mule deer age ratios (fawn:doe), followed by the proportion of premium high-quality habitat in summer and suitable habitat in winter. As multiple forage metrics influenced by temperature and precipitation explain habitat suitability, weather effects may be even stronger than our estimates suggest. Given recent and projected increases in temperatures and instabilities in winter weather, populations of mule deer in Wyoming may experience reduced recruitment over the next half-century. Preserving or enhancing habitat, such as managing invasive annual grasses or maintaining wild horse populations at appropriate management levels, could help buffer against the negative effects of rising summer temperatures in some areas.

Our resource selection analyses rely on the assumption that patterns of resource use estimated over the timespan of the GPS location information apply forwards and backwards through time, i.e., resource selection is static for mule deer. Mule deer have high fidelity to seasonal ranges, migration routes, and migration behaviors (migrant versus resident) relative to other ungulates[40,41]. This aligns with the critical assumption that resource selection patterns are static, but it also suggests mule deer may be more reliant on

seasonal habitat conditions than other ungulates and consequently more susceptible to climatic and anthropogenic changes.

Temperature, extreme cold and heat, negatively impacted mule deer age ratios at a rate 1.3 to 2.5 times greater than the positive influence of habitat suitability. By 2050, mean temperature in Wyoming, USA is expected to increase by 2.5–3.5 degrees C under RCP 4.5 and 8.5 emissions scenarios, respectively (based on 20 CMIP5 model mean[42]). Increasing summer temperatures can decrease the survival of neonates by influencing their body condition and that of their mother while nursing[28]. Mule deer pregnancy rates, though typically high (25-year mean = 98%[28]), depend on the doe's energy intake before and during breeding[26], thus temperature-related impacts resulting in negative energy balances entering the breeding season could potentially lower pregnancy rates. Temperature plays a role in forage quantity and quality patterns[43]. Thus, the full accounting of the effect of temperature may be larger than estimated given the inclusion of multiple forage covariates in the habitat suitability analysis. Furthermore, warming temperatures are projected to increase the frequency of drought by as much as 100% across much of North America by the end of the 21st century[44]. Extreme drought events can have multiplicative effects on fawn survival both through direct mechanisms such as heat stress and dehydration but also indirectly by influencing the quality, quantity, and availability of forage on the landscape, which can lead to starvation[24,45]. Given the adverse drought effects, adopting management practices that support soil moisture retention and drought-resistant perennials may be especially beneficial for mule deer.

**Fig. 3 | Seasonal habitat suitability and persistence in Wyoming.** Maps of average habitat suitability scores (**a**, **b**) and the number of years each pixel meets the relevant definition of habitat for summer (**c**) and winter (**d**) from 2000 to 2023 based on resource selection function (RSF) analyses of female mule deer in Wyoming, USA. Black lines in (**c**) and (**d**) represent herd unit boundaries. The background relief in (**a**) and (**c**) is based on the SRTM 90 m digital elevation model[67].

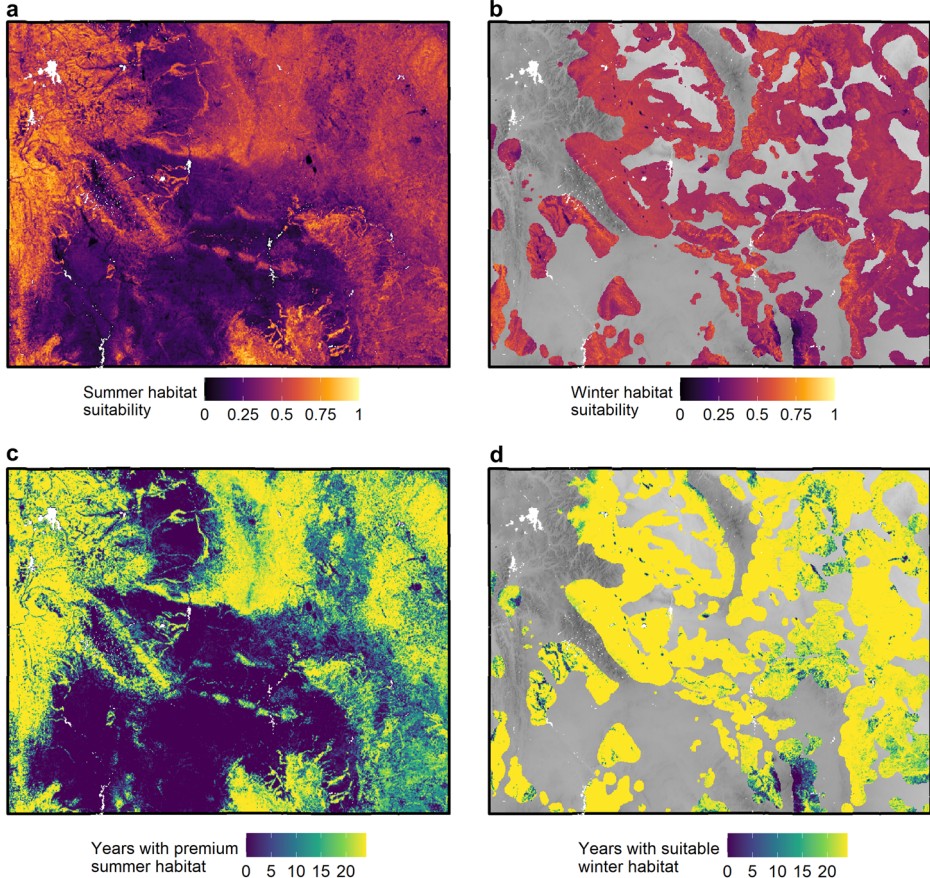

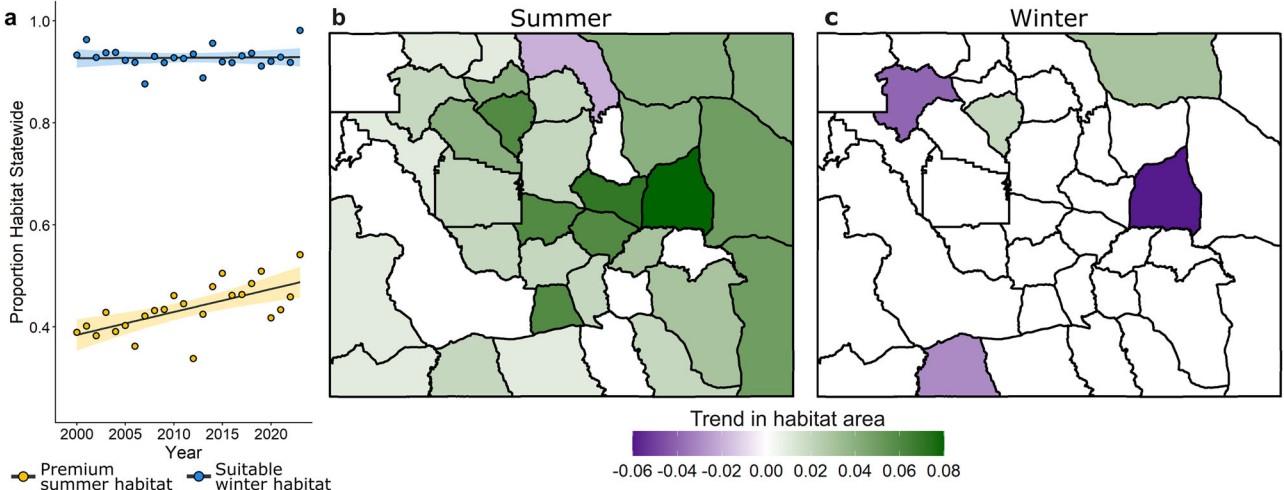

**Fig. 4 | Spatial and temporal trends in mule deer habitat in Wyoming.** Annual change in the amount of mule deer habitat (**a**), premium summer and suitable winter habitat, calculated for the state of Wyoming, 2000 to 2023. Spatially explicit trends in premium summer habitat (**b**) and suitable winter habitat area (**c**) across Wyoming mule deer herd units, 2000 to 2023.

Warmer winter conditions could bode well for recruitment during the following spring by increasing over-winter survival and adult female body condition during gestation[46]. However, warming global temperatures have led to a destabilization of Earth's atmospheric flows, which is hypothesized to be a mechanism for increases in the frequency of extreme cold and snowfall events in the Northern Hemisphere[47,48]. Such events have led to considerable winter die-offs in recent years[46], most notably in the Wyoming Range herd during the 2022–2023 winter when adult female deer survival fell to 33% (in contrast to ranging between 66% and 97% over the prior decade) and an estimated 92% loss in the 2023 fawn cohort[49]. Temperatures during this time were consistently below freezing with multiple daily minimums falling below −25°C. Above average snowfall occurring in 2022–2023 likely also exacerbated already harsh winter conditions. Age ratio surveys during the winter of 2024–2025 suggest reproductive recovery is possible after two years of mild winter conditions and higher summer fawn survival[50]. Despite signs

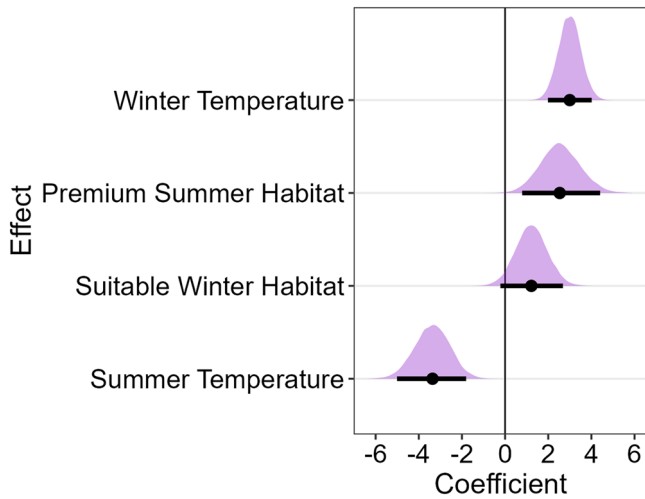

**Fig. 5 | Posterior estimates of environmental covariate effects on Mule Deer age ratios.** Model results relating environmental covariates to age ratios for mule deer in Wyoming, USA from 2001 to 2023. Full posterior distributions are represented along with mean effect (points) and 95% credible intervals (bold horizontal lines). Numerical representation of coefficients can be found in Supplementary Table 6.

of increases in reproductive success, mule deer abundance across Wyoming remains 50% below the 40-year average[51].

We were unable to include the effect of precipitation on mule deer age ratios due to its correlation with predicted habitat suitability ($r = 0.58$). However, this correlation suggests dietary and shelter components of habitat are regulated, in part, by precipitation. In our analysis this includes variables like NDVI, biomass and cover of perennial forbs and grasses among others. Contrasting with temperature, future projections of summer precipitation in Wyoming, USA are uncertain, although minor increases in winter precipitation are projected by 2050 (up to 25.40 mm[42]). Ultimately, our understanding of the effects of future temperature and precipitation regimes on mule deer will be advanced through studies that forecast habitat conditions across a range of climate scenarios, capturing the uncertainty in climate projections. Such research could enhance managers' abilities to proactively allocate resources to areas with greatest need.

In modeling RSFs our goal was not to redefine the important components that define mule deer habitat. Rather, we relied on the vast amount of research already conducted on this topic to build biologically relevant models and used these models to predict habitat suitability for mule deer through time across Wyoming, USA. Using the RSF predictions to explain demographics while accounting for the abiotic conditions experienced by a species provides a broader perspective on the relative influence of abiotic and biotic factors driving aspects of recruitment.

The amount of premium habitat available to mule deer during summer positively influenced age ratios during the subsequent winter. Premium habitat encompassed 76% of the locations of deer in this study, while covering 35–58% of actual land area depending on the year. Premium high-quality summer habitat is defined at a macro scale by landcover tied to forage most relied upon by mule deer during this period and productivity (amplitude and rate of green-up). These metrics broadly represent the quantity and quality of vegetation and are heavily influenced by precipitation. From a dietary perspective, premium summer habitat increases with higher amounts of forage like woody shrubs and perennials in areas of increased productivity (Fig. 2a). Availability of premium summer habitat varies across the study area (Fig. 3a). For example, migratory herds in the western half of Wyoming have access to consistently high-quality habitat in the Absaroka, Wind River, and Wyoming Ranges (Fig. 3a). Even with potential statewide gains in the amount of premium summer habitat there are areas where summer habitat is lower quality or less available. Resident populations east of the Bighorn Mountains in northeastern Wyoming exist

within a more heterogenous landscape of habitat suitability wherein some summer habitat is often less suitable (Fig. 3a), and in other areas of Wyoming, such as the southcentral portion, habitat suitability is more homogenous and consistently lower, rarely meeting the classification conditions of premium quality habitat during the study period (Fig. 3a). Given the pronounced spatiotemporal variability in summer habitat quality, targeted efforts to enhance and expand premium habitat, especially in areas of persistent limitation, may be critical for sustaining mule deer populations across Wyoming.

The amount of suitable winter habitat also positively influenced age ratios of populations in the following year. In temperate environments, female ungulates are tasked with balancing their own survival and devoting resources to fetal development in unpredictable weather conditions during winter. The availability of suitable winter habitat on the landscape hinges not only on the amount of winter forage, e.g., sagebrush cover and herbaceous biomass remaining from the growing season (Fig. 2b), but also on the accessibility of said forage which can be limited by snowfall in extreme winter conditions. The importance of winter habitat in supporting mule deer populations aligns with the focus on conservation actions in recent years to improve habitat quality on winter ranges of western big game (e.g., Secretarial Order 3362, U.S. Department of the Interior[52]). We found that within areas predominantly used by mule deer during winter, where management actions are often directed, suitable winter habitat remains prevalent and consistently available across the years of our study (Fig. 3d).

We found that mule deer avoided areas with more energy infrastructure, aligning with past studies that showed an avoidance of roads, particularly during installation when worker activity was highest[18]. The intensity and consistency of traffic on roads, data that do not exist at the scale of our analysis, may more likely be the ultimate driver behind mule deer avoidance of roads[19,53], rather than the presence of roads themselves. We only observed strong road avoidance in summer in the western region of Wyoming, an area that experiences an influx of millions of tourists annually, leading to increases in traffic, though we were unable to test this explicit connection in our analysis. We found no support for a road effect on winter habitat selection. We hypothesize this result is due to selection of home ranges with minimal roads and structure of the data rather than an absence of avoidance behavior. Third-order selection with single daily locations may also mask fine-scale movement and subsequent avoidance of roads, especially in winter.

Detailed information on predation risk was not available at the scale of our study and is absent from the resource selection analyses. Nonetheless, some covariates used to model habitat suitability, such as topography and vegetation cover, may serve as proxies for spatial variation in predation risk. Future research could benefit from explicitly incorporating predator density or predator-specific resource selection patterns to more fully capture the influence of predation on mule deer habitat use.

From a management perspective, supporting mule deer populations by understanding the drivers of demographic rates is a multifaceted issue. While seasonal temperatures are outside of the levers of management, we have identified key classifications of habitat where conservation actions could prove beneficial. Potential high value priorities include continuing to build on efforts to improve habitat quality in summer, with particular emphasis on temperature refugia and forage robust to drought. Mule deer populations in northeast and southcentral Wyoming may particularly benefit from increased attention. Preserving the current relatively high integrity of habitat in the identified winter use areas will also support higher fawn:doe ratios. Management actions that align with these priorities include invasive annual grass and conifer encroachment treatments which are working to restore core sagebrush ecosystems[54].

Our findings also reaffirm the broader literature regarding avoidance of areas with high densities of oil and gas infrastructure[18,19,55,56]. By delineating areas of high habitat suitability and availability across space and time, our results provide spatial tools to inform siting or mitigation considerations for energy development or other infrastructure proposals in mule deer summer and winter use areas in Wyoming. The methods and procedures described

herein could be implemented to inform similar assessments in other areas of the western United States. This study is part of the Wyoming Landscape Conservation Initiative and is supporting the development of multispecies conservation prioritization tools while serving as a blueprint for research on other ungulate species.

## Methods

### Data overview

We compiled previously collected GPS-collar data from large numbers of adult female mule deer across the state of Wyoming, USA (Fig. 1). These data represent information from 21 study herds generated over a 22-year period, from 2001 to 2022, and have been used in many previous studies on mule deer in this region (cited throughout this manuscript). No new data collection was conducted for this study and all data within was compiled from studies that complied with all relevant ethical regulations for animal use. There was an average of 70 (range 14–295) individuals monitored from each study herd, a 449-day average monitoring period and fix rates ranging from one to 16 relocations daily. The home ranges of these study herds are defined by diverse landscape characteristics including elevational gradients, variable vegetation and faunal assemblages, land ownership patterns, and energy development. These diverse landscapes include the conifer-dominated forests of the Southern and Middle Rockies, the sagebrush shrublands and semi-arid deserts of the Wyoming Basin, and the rolling lands of the Northwest Great Plains[57].

### Informing habitat suitability via resource selection functions

We modeled third-order (within home-range) resource selection on summer and winter use areas with a used-available design[58]. We segmented the GPS-collar data into animal-years, a combination of unique animal identity and biological year beginning on July 1. Seasonal use areas for migratory individuals were defined on an individual basis as follows: summer was the period between July 1 and the fall migration start date, winter was the period between the fall migration end date and the spring migration start. We delineated the start and end of each migration by manually inspecting net squared displacement curves[59] and maps of the GPS locations using the Migration Mapper application[60], and program R. We classified locations of non-migratory individuals as being in winter use areas based upon the median migration dates of all migratory individuals resulting in a winter period for non-migrants between November 17th and May 22nd. We chose this approach because nearly a third of study herds ($n = 6$) consist of a majority non-migratory individuals ( > 60%, Supplementary Table 1). We estimated summer and winter home ranges for each animal-year as the 99% isopleth of a kernel density estimator[61]. We thinned GPS data to one random location per day to define the sample of used locations for each animal-year. Data thinning limits temporal autocorrelation in used locations which can bias resource selection analyses[62]. We defined availability with a 1:1 used:available ratio and sampled the available set within each individual deer's summer and winter home range. We conducted a priori testing of beta coefficient stability at different used:available ratios (1:1, 1:5, 1:10) and found nearly identical results across ratios indicating our choice of a 1:1 used:available ratio was robust given our large dataset while reducing the run time of models from days to hours.

We used the Rangeland Analysis Platform (RAP) to obtain the fractional cover of annual forbs and grasses, perennial forbs and grasses, and shrubs[63]. Forage resources can be a critical component of seasonal mule deer resource selection; therefore, we separated sagebrush from the general shrub layer to account for the seasonal variation in use of these cover types. We generated a non-sagebrush shrub layer by subtracting sagebrush-specific shrub cover (*Artemisia* sp.) defined by the Rangeland Condition Monitoring Assessment and Projection project (Rigge et al. 2024) from the RAP shrub layer. We used Normalized Difference Vegetation Index (NDVI) and instantaneous rate of green-up (IRG) as proxies for the quantity and quality of ungulate forage[38,64]. For each calendar year, we included covariates for the amplitude of the NDVI time series as an index of maximum greenness, the peak IRG, and the Julian day of peak IRG. Amplitude was derived from a

MODIS NDVI product (MOD13Q1) and processed IRG data, generated from MODIS (MOD09Q1), was retrieved from Wildlife Move Tools data repository (www.wildlifemovetools.org). We also derived the total above-ground herbaceous biomass from RAP for use as a proxy of forage availability at the onset of winter[63].

Mule deer use certain terrain features to reduce predation risk, thermoregulate and locate forage[16,17]. We calculated three metrics to represent the topographic landscape: heat load index[65], terrain wetness index[66], and terrain ruggedness index based on the standard deviation of the curvature using a 90 m digital elevation model for the state of Wyoming[67,68]. We also used mean snow water equivalent (SWE[69]), to inform variation in winter selection that may be driven by annual differences in snow cover. Human development impacts habitat selection in mule deer[18,19,55,56], and we used a time series of oil and gas wells in Wyoming developed by Hayes et al.[70] to represent annual density of energy infrastructure (per km$^2$). We also considered the influence of wind energy development on mule deer resource selection, however too few deer in our dataset ($n = 3$) had home ranges including wind turbines. We created distance to primary and secondary roads[71] and permanent water body layers[72] based on negative exponential decay functions where the effect declines to zero at 500 m, to approximate the spatial impact of these features on habitat selection. All spatial data processing and analyses were conducted using package terra v.1.8.42[73] and whitebox v.2.2[66] in program R v.4.4.1[74]. Explanatory covariate data layers were scaled to 90 m resolution, and we extracted the pixel values underlying the used/available locations matched to the biological year of said locations.

We predicted the strength and direction of selection may differ between geographically distant subpopulations because of the landscape diversity and resource heterogeneity experienced by mule deer across Wyoming. Therefore, we modeled resource selection for seven region-season combinations (three summer and four winter regional groups, based upon geographic distance and ecological similarity, Fig. 1).

We estimated the resource selection functions using a logistic regression framework. Recent work on the presence and importance of among-individual heterogeneity in resource selection patterns has underscored the need to incorporate individual-level random effects in resource selection functions to improve population-level inference[75]. To account for this, we used a hierarchical mixed-model structure (i.e., a combination of fixed and random effects). We fit global models with all explanatory covariates for all season and region combinations after evaluating multicollinearity among covariates (all variables met conditions of $r < 0.60$). We modeled resource selection ($\pi_{nj}$), the probability that a point $y_{nj}$ with $X$ covariates is used, for $N$ individuals at a set of $J$ locations with this general formulation:

$$y_{nj} \sim Bern\left(\pi_{nj}\right), logit\left(\pi_{nj}\right) \sim \beta_{0n} + \beta_{xn} * x_j$$

where $\beta_{0n}$ is an individual-level random intercept and $\beta_{xn}$ represents individual level random slopes for a set of $X$ covariates. Individual-level effects were drawn from a set of hyperparameter distributions representing region-level mean effects:

$$\beta_{xn} \sim Norm(\mu_x, \sigma_x^2)$$

with $\mu_x$ representing regional mean effect with some variance $\sigma_x^2$ among individuals. Hyperparameters were modelled with vague prior distributions for population means (e.g., $\mu_x \sim Normal(0, \sigma^2)$ where $\sigma^2 \sim Gamma(1, 1)$). Explanatory variables were standardized with $\bar{x} = 0$ and $\sigma = 1$ to allow for direct comparison of effect sizes. We fit all models in a Bayesian framework using NIMBLE v.1.3[76,77] in program R v.4.4.1[74]. Each model ran for 35000 iterations over four chains with a thinning rate of five and a 5000-iteration burn-in leaving 24000 samples for inference. Chain mixing was checked by visual assessment and Gelman-Rubin statistic ($\hat{R}$). We conducted k-fold cross-validation, partitioning data into five equal-sized folds, ensuring each fold is a random subsample of individuals to maintain independence between training and validation data[78]. Predictive performance was assessed

by ranking RSF scores from the validation fold, binning them into habitat suitability levels, adjusting frequencies by availability, and computing the Spearman rank correlation between bin ranks and area-adjusted frequencies[78]. Spearman rank correlation for frequency values by bins indicates that the models predicted cross-validated use locations well (summer: $r = 0.91$, $P < 0.01$; winter: $r = 0.85$, $P = 0.03$).

Preliminary assessment of GPS locations indicated many individuals did not encounter oil and gas development within their home ranges (75% in winter and 68% in summer). Such lack of availability of a given resource can induce bias in RSF modeling. Therefore, we pooled individuals that encountered energy development across regions under the assumption that response to this would be similar. We fit univariate models to determine the effect of energy development on selection for winter and summer periods and multivariate (all RSF covariates) models to evaluate collinearity with other covariates. The estimated effect of energy differed little between univariate and multivariate models therefore we used univariate estimates in predictions.

## Estimating changes in habitat suitability over time

Our goal was to link patterns of resource selection to a time series of age ratios for mule deer across Wyoming. We used the estimated resource selection functions to predict habitat suitability in summer and winter from 2000 to 2023 by applying the estimated equations to the time series of covariates to estimate the relative probability of selection for each pixel in each year and standardized selection values between 0 and 1. Only variables with strong statistical support, 95% CRIs not overlapping zero, were used to produce the selection surfaces. We generated predicted surfaces of habitat suitability across Wyoming from 2000 to 2023 based on the resource selection function (RSF) results from each modeled combination of season and region. We completed a priori checks on the alignment of habitat covariates in- and outside of individual home ranges and found broad overlap indicating the large spatial extent of global positioning system (GPS)-collar data used in this study captures the range of habitat conditions statewide. However, there are portions of Wyoming where mule deer live but from which we have no GPS data to inform our RSF analyses. To interpolate such that predictions were more heavily influenced by models from deer in closer proximity, we chose an inverse distance weighting approach where weights declined with distance. We then produced a single aggregated layer of habitat suitability for each year and season combination from the suite of inverse distance weighted layers. The spatial weighting process for each region and season combination is described in Supplementary Fig. 1.

The weighted averages of regional selection produced a single predicted probability of selection layer for each year and season combination. Using the continuous surface predictions, we derived annual season-specific discrete measures of habitat quality. For each used location, we extracted the predicted suitability values for the corresponding year. We calculated two cutoffs based on normal probability distributions to categorize pixels as premium and suitable habitat[79]. Premium habitat included all values greater than 0.5 σ below $\bar{x}$ ( ~ 40th percentile) and suitable habitat included all values greater than 1.5 σ below $\bar{x}$ ( ~ 10th percentile).

## Relating age ratios to environmental covariates

We used estimates of age ratios from 2001 to 2023 across 34 herd units generated from annual surveys by Wyoming Game and Fish Department (WGFD) which predominantly occur in December[80–82]. To understand the biotic and abiotic drivers of age ratios, we modeled age ratios in year $t$ as a function of the proportion of suitable and premium summer and winter habitat, winter severity, and drought conditions. All winter covariates were measured in year $t$ -1, coinciding with the gestation period for maternal females counted the following winter. Summer covariates were measured in year $t$, representing conditions that affect fawn survival in the time prior to counts. We calculated average winter severity and drought metrics from December through March and June through August, respectively. We evaluated several indices of winter severity and

drought conditions. Winter severity metrics included basic winter severity, accumulated winter severity, SWE, average minimum temperature, and total precipitation. We generated a basic winter severity index by calculating the difference between standardized ($\bar{x} = 0$, $\sigma = 1$) total precipitation and average temperature across the entire winter season[83]. Accumulated winter severity (AWSI) indexed winter severity by summing the number of days during the winter season with measurable snow cover or average temperatures below freezing. We calculated AWSI using data from MODIS (MOD11A1 and MOD10A1) in Google Earth Engine[84]. Drought condition metrics included: the total summer precipitation and average summer temperature. During preliminary analysis we also considered the inclusion of two other drought metrics, the number of consecutive prior months of severe drought and average daily drought severity, based on the self-calibrated Palmer Drought Severity Index (scPDSI)[82,85]. However, metrics based on scPDSI lacked sufficient annual variation to estimate the influence on age ratios and potentially indicate a new normal of prolonged periods of drought across the study area. For any given year, the average herd unit was in drought status (scPDSI < 0) and experienced 9.5 months of severe drought (scPDSI < −3, range = 0–45 months) in the previous five years. Unless otherwise noted, seasonal precipitation and temperature data were derived from PRISM[86].

Relating explanatory covariates to age ratios at the scale of the WGFD herd units involved additional steps to ensure appropriate spatiotemporal alignment with areas used by deer. In some regions of Wyoming, portions of mule deer populations migrate during spring and spend summer on herd units adjacent to their wintering areas. Thus, we needed to measure summer conditions where mule deer spent time, to correctly gauge the effects of summer conditions on age ratios (estimated on wintering grounds). We determined the proportion of deer that summer in adjacent herd units and those that stay within their wintering herd unit using GPS data and calculated a weighted average of the proportion of suitable habitat across used herd units (e.g., Supplementary Fig. 2). Winter covariates were summarized across winter use areas of mule deer within each herd unit, defined by GPS location data (Supplementary Fig. 3), allowing for direct assessment of the effects of winter conditions in the portion of the herd units where age ratios are determined.

We modeled the herd unit-specific time series of recruitment ratios in a Bayesian mixed model framework with the brms package v.2.2[87] in program R v.4.4.1[74]. In all models we included a random intercept term for herd unit. Explanatory covariates were categorized into four distinct sets representing summer and winter habitat, winter severity, and summer abiotic conditions (Supplementary Table 4). Due to high multicollinearity within some variable categories, we used an iterative model selection process beginning with fitting univariate models for all prospective variables and retaining those with the largest effect size from each category[88]. We then modelled all combinations of variables across categories (Supplementary Table 5) and selected the best fitting combinations using a Bayesian leave-one-out cross-validation (LOO) estimate of out-of-sample predictive fit which generates an information criterion (looic[89]), asymptotically equal to the more commonly used Watanabe-Akaike information criterion (WAIC). As WAIC can be a less robust estimate of out-of-sample predictive power, we used LOO as the suggested alternative[87,89]. Models ran for 35,000 iterations over four chains with a thinning rate of five and a 5000-iteration warm-up leaving 24,000 samples for inference.

## Reporting summary

Further information on research design is available in the Nature Portfolio Reporting Summary linked to this article.

## Data availability

Source datasets for resource selection and age ratio models and seasonal habitat suitability raster data are available[35]. Raw mule deer GPS location

data used in this study is part of a broad collaboration with many partners and interest in access to this information should be relayed to the corresponding author.

## Code availability

A description of the resource selection model formulated for use in NIMBLE is available in the Supplementary Information. No additional novel code was developed for these analyses; all software packages, versions, and programs used are documented within.

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

## Acknowledgements

Funding for this study was provided by the U.S. Geological Survey Wyoming Landscape Conservation Initiative, Species Management and Biothreats programs. We thank the Wyoming Game and Fish Department for in-kind support along with many others who supported this project through mule deer data acquisition and contributions during the planning stages (Justin Binfet, Todd Cornish, Teal Cufaude, Melia Devivo, Sam Dwinnell, Gary Fraylick, Pat Hnilicka, Rusty Kaiser, Lee Knox, J. Terril Patterson, Erika Peckham, Jill Randall, Hall Sawyer, Jeff Short, Cheyenne Stewart, Tim Thomas, Mark Thonhoff, Brandon Werner). We also thank the Wind River Inter-Tribal Council for sharing their mule deer data. Any use of trade, firm, or product names is for descriptive purposes only and does not imply endorsement by the U.S. Government. The findings and conclusions in this article are those of the author(s) and do not necessarily represent the views of the Bureau of Land Management.

## Author contributions

W.M.J., A.N.J., S.L.B., S.R.D., K.S.H., T.N.L., B.L., R.P.J., T.A.G., E.H., T.A.H., M.J.K., and K.M. contributed to the conceptual development of the study, devised analytical approaches, and supported data acquisition. W.M.J. and T.A.G. analyzed the data and wrote the first draft. W.M.J., A.N.J., S.L.B., S.R.D., K.S.H., T.N.L., B.L., R.P.J., T.A.G., E.H., T.A.H., M.J.K., and K.M. were involved in interpretation of results, provided input on figures, revised multiple manuscript drafts, and approved the final submitted version.

## Competing interests

The authors declare no competing interests.
