## [Transparent Peer Review file · Communications Biology]

The interplay of habitat quality and temperature shape demographic patterns of mule deer (*Odocoileus hemionus*) in North America.

Corresponding Author: Dr William Janousek

Version 0:

Reviewer comments:

Reviewer #1

(Remarks to the Author)

This was a well written and innovative paper that was interesting and enjoyable to read. The author's approach is innovative, and the strength of this paper lies in the large spatial scale, the ability to account for variation over time at that scale, and the link between habitat and abiotic factors to the age ratios/recruitment for mule deer. In my opinion this will contribute to the future of ungulate research in general, as well as mule deer research and management in particular. The analysis was robust and well thought out, and while I have one comment on the analysis, most of my comments are about framing and interpretation and are relatively minor. I look forward to seeing this published.

Major comment:

The only concern I have about the analysis is the correlation between the winter and summer abiotic variables used in the age-ratio model and the variables used in the RSF to generate the suitable and premium habitat layers. Summer precipitation and temperature are likely to be correlated with the NDVI and the peak green-up metrics, and the authors even found that summer precipitation was correlated with the entire habitat layer. I mention this because higher summer temperatures may have an additive effect of both shrinking the amount of premium habitat and causing heat stress and other heat related issues, as mentioned by the authors on lines 208-211 in the context of extreme drought. I think the authors need to mention this possible correlation in the methods as well as the discussion in the context that the effect of temperature may be even larger than found by the study. I would also recommend that the authors show that temperature is not correlated with the NDVI metrics used in the RSF analysis.

Minor Comments:

Lines 63-65: This sentence is unclear to me. Are you saying that there is strong concern that abiotic factors will degrade habitat for ungulates? If so state that more clearly. If you are stating that habitat quality (and by that I assume you mean forage quality) is strongly linked to ungulate dynamics and so any declines in forage/habitat quality are concerning for the management of abundant ungulate populations I would state that more clearly.

Line 73 – I would say management concern because there is little chance of actual extirpation of mule deer populations, but opinions may differ

Lines 104-105 – This sentence highlights the issue I raised in the comment above, that winter severity and summer temperature/precipitation are inextricably linked to the quality of the habitat. This requires a bit more acknowledgement by the authors that these two are linked and the abiotic may influence that amount of available habitat, but I still think the analysis approach is robust

Line 129 - There were some exceptions to the concept of "different by region but similar in direction" in the sagebrush (trend for positive in Northeast) and heat load index (trend for negative in Northeast) variables that were not addressed in the result text

Lines 146-149 – This is definitely a pretty moderate correlation for this validation, probably because of the large variation between the regions, but I would mention this variation more than claim that this moderate correlation is indicative of consistent results and instead say that even with the noise in the models from regional variation it was still an informative covariate for the age-ratio models

Line 159-160 – very interesting result and one that informs the debate on the importance of summer vs winter habitat

Lines 174-178 – I would lead with lines 177-178 and then discuss the magnitude of summer temp to make it easier to follow the findings and see the overall magnitude and importance of summer temp

Lines 205-206 – Given the importance of summer temperature in the model results I recommend adding another sentence and expanding possible heat stress impacts on recruitment

Lines 220-225 – If there is any information on how long deer herds take to recover from a winter event of this magnitude it would good to include here. I think some of the focus on winter range vs summer range is because severe winters have a major and visible impact on mule deer herds while poor summer habitat leads to more gradual declines

Informing management of mule deer – It may be beneficial to more explicitly recommend that if agencies have limited budget for habitat restoration or management to focus on summer range. Since winter range was consistent across years monitoring winter habitat while restoring summer habitat seems like it would be better to slow mule deer decline. The results of this paper also indicate, to me at least, that summer habitat restoration will only slow the decline and it would be hard pressed to reverse the decline if summer temperature is the strongest ecological driver of recruitment.

Line 301 – A word seems to be missing here “The ____ of these study herds...”

Lines 357-358 – Add slightly more information here so this is repeatable. Did the effect decay to 0 at 500m? Was this linear decay or another type of decay function?

Lines 373-374 – Table S2 referenced in the text is not a table of models but is actually the selection coefficients for covariates, as correctly referenced on line 130. The table with the a priori models needs to be added to the Appendix

Nice work on the figures. Clean, clear, and informative

Reviewer #2

(Remarks to the Author)

Overall, I commend the authors on what will no doubt be an important contribution to the field of ecology as well as the management and conservation of ungulate populations. I found the study’s stepwise progression, from assessing mule deer resource selection, to using those selection results to measure habitat suitability, to finally linking habitat suitability and other environmental variables to fawn:doe ratios, to be both a creative and intuitive approach. The fact that this study was also done with such an extensive dataset, spanning almost 1500 mule deer over 22 years from multiple herds across Wyoming, makes it even more impressive. Furthermore, the analytical approaches used to conduct this study were, in my eyes, statistically sound and defensible. This section of the manuscript was also very strong and well written, with a clear and comprehensive overview of the methods. Finally, the tables and figures throughout the manuscript and SI were informative and aesthetically strong. It was clear that significant thought, time, and effort went into producing them.

With that being said, I was a bit disappointed with the quality of the writing throughout parts of the main text, especially throughout the Introduction and sections of the Discussion. Currently, the main text is rife with run-on sentences that attempt to string together too many ideas. These run-ons, combined with cases of poor sentence structure and transitions, made it difficult to follow the manuscript at times. Without considerable revisions to the writing, the main text does a disservice to all the fantastic work the authors put in to amassing such an impressive dataset and conducting what I believe to be an interesting and broadly impactful study.

As a result, I offer many line-by-line comments below, most of which apply to textual edits, in the hopes that they improve the quality of writing in the manuscript.

Abstract

L42-44: Consider changing sentence to following: “Identifying drivers of mule deer demography could inform habitat restoration. However, linking habitat quality with variation in vital rates is challenging and often done indirectly through proxy metrics.”

L45: Replace “We apply a novel approach combining...” with “We combine...” to avoid using qualifiers like new and novel as the journal suggests. Also cuts down on words to get you closer to the 200-word limit (currently at 209) if necessary.

L46: Minor, but I would consider just using fawn:doe ratios throughout instead of age ratios. Then readers don’t have the make the extra jump each time to remember age ratios is only referring to fawn:doe ratios, as opposed to all variation in age within populations.

L46: replace “data of” with “data from”

L48: replace “then modeled” with “model” to maintain tense of rest of sentence.

L51-57: Nice closing to your abstract that highlights the impact of your work, as well as it's only growing importance in future decades! Consider some minor suggestions to improve the flow of the last sentence: "Our findings also show mule deer avoid areas with high densities of oil and gas extraction. By delineating important mule deer habitat, we offer spatial tools for development siting and mitigation across the western United States."

Introduction

Overall, I found the introduction difficult to get through. It contains valuable information pertinent to the research objective, but most of it is overwhelmed by run-on sentences that attempt to convey too many points at once. As a result, the introduction lacks a cohesive flow and fails to draw the reader in. Most of the introduction would benefit from a focused effort to split longer sentences into multiple, and streamline language to improve brevity while also taking the time to provide clarity and context for the reader. I offer some examples below of how this could be done:

L63-65: This sentence is quite vague and feels repetitive. Not really understanding what your message is. Maybe this better conveys what you're trying to say?: "Abiotic factors are strongly connected to the biotic world. Changing climates and landscapes lead to habitat alteration, which in turn effects ungulate populations that depend on said habitat. These connections motivate current concerns over the conservation of ungulate populations amidst changing habitat."

L65-70: Wonderful summation of the major ways in which changing habitats are affecting ungulate populations.

L71-74: Since you have plenty of words to work with, you might consider splitting this into multiple sentences. Because the paper focuses on mule deer and jumps into them fast, I think readers would appreciate a couple of sentences going into more depth on how they are economically and ecologically important. Highlight some of what's discussed in the two papers you cite here. Then you can transition into their decline with something like "However, mule deer are experiencing declining numbers across their range due to..." And assuming the reason is habitat change, you could then add how important it is to accurately quantify what habitat means for mule deer and end the paragraph with that. This would segue into the following section nicely where you discuss attempts to identify characteristics of mule deer habitat.

L74-79: Move comma after "White et al 2025)" to after "across scales and seasons"

L82-83: Opening to the sentence feels clunky, consider "While this complexity makes it hard to define "suitable habitat" for mule deer, seasonal..."

L89-92: This sentence feels fragmented and disjointed to where I'm not entirely sure what you're trying to say. What is meant by "habitat selection indicative of security by the mother during early life" and how does this relate to the rest of the sentence and the point made in the prior sentence? I suggest rewriting this sentence to flow more logically, by first mentioning how habitat selection has been shown to influence body condition, and then how body condition has been connected to variation in vital rates.

L100-114: This paragraph does a nice job succinctly summing up your study. Well done! However, because your methods come at the end of the paper, I think you need to move some key details either to this paragraph or within the "Results" section (this might be a more natural fit). For instance, I wasn't aware that 1) fawn:doe ratios were collected at the start of winter in December, and 2) when you say winter habitat and temperature, you're referring to winter t-1 during doe gestation, rather than the first winter experienced by the fawn. It should be easy to weave this in earlier than methods.

Results

Great results section overall. Succinct and clearly describes the key findings of the study. Just a few minor suggestions below.

L117: Since this follows the "Methods at the end" structure, make sure to define any acronyms mentioned for the first time in your Results/Discussion rather than in your Methods. For example, define GPS here instead of in Methods.

L127: Again, because Methods come at the end, I would open this and subsequent sections with a quick intro sentence stating what was done. Just to prime the reader for the results they're about to look at. Here, I would open with a quick list of the covariates you considered. One option: "We examined mule deer resource selection of various landscape covariates deemed a priori to be important. In summer, this included... In winter, this included..."

L133: Consider adding a few citations after "aligns with the literature" to support your claim.

L145-149: Consider restructuring this long sentence and breaking it into two. Something like this "By using the conservative approach of randomly subsetting data into folds without respect to individual identity and used-available status, cross-validation scores indicated moderate correlation among folds (0.51-0.56). Thus, demonstrating that our models have enough statistical power to overcome such limitations and produce consistent results."

L150: add "showed" or a similar word in between "GPS data" and "that most". You can also remove the hyphen from GPS data.

L159-160: Since you already say "higher variability" at the start of the sentence, you can keep it simple and just say "in

summer than in winter,” as it’s already implied winter was less variable.

L161: replace “is” with “was” to be consistent with tense.

L159-165: While your use of premium for summer and suitable for winter makes sense to me for the purpose of your analyses, is it fair to also apply this to your discussions about seasonal variability in suitable habitat? In other words, is it defensible to compare differences in variability when you’re using a different quality threshold across seasons? For example, I would think the higher quality “premium habitat” would inherently be more variable than the lower quality “suitable habitat,” especially if meeting this “suitable” designation includes higher quality habitat designations like premium, since that would also be considered suitable. On that point, was any habitat classified as “premium” in winter? If so, it might be interesting to see that variability compared with summer or simply compare suitable with suitable. I also acknowledge that this is a very minor concern, as I assume the “suitable” winter designation is roughly equivalent to the “premium” summer designation, since habitat suitability is expected to be much lower overall in winter.

L167-169: Another place where you might consider clarifying that winter habitat and temperature

L171: Briefly define “looc” here since this is the first mention

Discussion

Aside from some suggestions to improve the clarity of the writing, I think you have a nice discussion that hits most of the key points of the study and how it extends to the management/conservation of mule deer. With that being said, since you didn’t factor in predation risk or overwinter survival and true recruitment in your analyses, it would be beneficial to devote a paragraph to these topics/limitations of your study (i.e., mentioning the importance of predation/predation risk for shaping ungulate movements and survival, especially for juveniles; any thoughts on how future work could be improved by trying to incorporate predation/risk; importance of measuring overwinter survival and true recruitment for understanding impacts of habitat and resource selection on population demographics, what do your significant results about winter t-1 say about how strong of an effect the current winter may have on recruitment/survival?).

L180-184: This sentence feels too long and has too much going on. I would recommend finding a way to split this into two sentences.

L184: Remove novel approach to align with journal guidelines. Could instead say “By combining habitat selection with climate-related effects, we identify...” In a similar vein, while I recognize that it is important to showcase how your study has moved the bar forward (and I agree that it has done a nice job of this), I don’t think this needs to be pushed as much as it currently is in the manuscript. For example, at several points you discuss that previous studies have only ever made indirect links between habitat quality/selection and vital rates/population dynamics or have only used proxy metrics. I would generally refrain from making these types of claims given the vastness of the literature. Undoubtedly, this link has been made somewhere on some scale. More so, even though your connection of habitat selection to fawn:doe ratios is getting closer to linking habitat to true demography/vital rates, fawn:doe ratios are still just a proxy for a vital rate, especially since you aren’t capturing overwinter survival or accounting for pregnancy rates within herds. In my opinion, I would just let your amazing data speak for itself and focus on all the relevant findings.

L190-192: This sentence is also a bit clunky. Consider this revision to improve flow: “However, preserving or enhancing habitat—such as by managing invasive annual grasses—could help buffer against the negative effects of rising summer temperatures.”

L193-195: The transition here is pretty poor and, as a result, feels random and out of place. Maybe this works better? “Another growing concern is that mule deer have high fidelity to seasonal ranges and migration routes relative to other ungulates (Morrison et al. 2021). While this aligns with the critical assumption that resource selection patterns from RSF analyses are static, it also means mule deer are more susceptible to climatic and anthropogenic changes.”

L197-200: This sentence is redundant to your earlier sentence describing high fidelity to seasonal ranges and migration routes. I would recommend removing it and ending on your point that this makes mule deer more susceptible to change.

L201-225: While this is a great section that adds a lot of relevant information on how changing weather and climate is expected to impact mule deer recruitment, I think it would benefit from placing all this in the context of your results. For example, if summer temperature and drought are expected to increase in coming years which should negatively impact mule deer based on these studies. How do your findings in this study directly support these claims? Always try and relate back to your own study throughout each discussion section.

L211-214: Another run-on sentence that’s difficult to follow. Consider this revision: “Given these adverse drought effects that place pressure on higher elevations to provide refugia from summer heat, adopting management practices that support soil moisture retention and drought-resistant perennials may be especially beneficial for mule deer.”

L218: Add a comma after “atmospheric flows” and just say “which is a hypothesized mechanism for...”

L230-233: Remove “is a novel approach and...” Also, while it’s certainly affecting recruitment, you might caution against using the term when applying to your study, since you only really capture an index of fawn survival up to but not including

winter (which can be pretty low depending on winter severity), and thus, aren't measuring true recruitment into the population.

L261-263: I would edit "We found avoidance of areas with increased amounts of energy infrastructure by mule deer and past studies have also shown an avoidance of roads particularly during the installation of energy development when worker activity is highest (e.g., Sawyer et al 2006)." in the following way: "We found that mule deer avoided areas with more energy infrastructure, aligning with past studies that showed an avoidance of roads, particularly during the installation of energy infrastructure when worker activity was highest (e.g., Sawyer et al 2006)."

L266-269: commas after "annually" and "traffic"

L271-273: hyphenate "third order" and "fine scale"

L281-282: simply saying "support demographics" is vague. I would replace with something like "support higher fawn:doe ratios."

L284-286: You might replace "oil and gas" with "infrastructure for oil and gas extraction" just to be clear what is meant.

L287: comma after "space and time"

In the last paragraph, you might consider adding a few sentences about the broader implications of your work for ungulate conservation. In other words, do your findings about mule deer across Wyoming apply more broadly to their entire range across western NA, can they shed light on how we might expect other ungulate species to respond with increasing global change? Just some thoughts for highlighting the broader impact of this great study!

Methods

Fantastic methods section. With great clarity and organization, you provided a very comprehensive overview of your analytical approach. Well done! Below are a few minor text edits and suggestions.

L301: remove "of these"

L303-304: hyphenate "conifer dominated"

L314: since you only use this once in the manuscript, I would replace "NSD" with "Net Squared Displacement" to save unfamiliar readers the trouble of figuring out what this means.

L324: typo, replace "availably" with "availability" or "available" depending on intention.

L336: replace "as proxies for forage" with "as proxies for the"

L360: hyphenate "90 m"

L409-411: I might just be reading this incorrectly, but if these thresholds are below \bar{x} , wouldn't it be more accurate to describe these as the 40th and 10th percentiles for defining premium and suitable habitat, respectively? Also, it might be worth moving these details to the results section so readers know how you are defining suitable and premium habitat, since this decision forms the backbone of the study.

L416-421: Here's where some of your details about the timing of summer and winter covariates could be moved into the main text to orient readers.

L434-438: Interesting (and sobering) little piece of information here that you gleaned from looking into this covariate that was ultimately dropped!

L463: replace "can be less" with "can be a less"

Tables and Figures

Figure 2: Consider adding summer and winter labels with figure letters as you did in Figure S1

Supplementary Information

Great work on the SI. Figures and tables are very strong. Beautiful while providing all the necessary information to support the main text.

Figure S2: If it's not too difficult, can you change the colors of the herds in the left plot to match the colors in the right plot? Also, there seems to be a mismatch between the data in the left plot and the right plot. The percentages on the right only add up to 93%, whereas on the left they appear to add up to 100%. Also, your statement that 76% summer in areas with higher proportions of suitable habitat suggests the Upper Shoshone value should be 24% rather than 17%, yet this wouldn't match

with the current distribution on the left plot. Instead, it looks like Dubois should be more like 40%, in which case the 76% should be 83%.

Reviewer #3

(Remarks to the Author)

The authors conducted resource selection analyses for a large dataset of mule deer across Wyoming and then correlated predicted maps of seasonal habitat, along with metrics of drought and winter severity, to December recruitment ratios. I found the manuscript to be very well-written and the work to be thorough and carefully crafted. I have a few questions geared at methods that might warrant further consideration prior to publication as well as some minor editorial suggestions.

42-43. I found this 2nd sentence of the Abstract to be a bit too big a leap, requiring some more detail or setup to jump to these complex themes such as demographic drivers of decline, habitat improvement projects, and the challenges of linking metrics of habitat quality to vital rates.

55. The phrase "oil and gas areas" is a bit vague.

136. The authors excluded migratory paths in their delineation of seasonal ranges, which I think negates their direct relevance to surfing of green waves (which happens during migration) despite migrating to areas with later peak productivity.

146. I'm unclear how to interpret these cross-validation scores nor is enough detail provided in the Methods to do so. Also, generally I find randomly sub-setting data by individual to be more robust in delineating independent training and testing folds of data than to create subsets without respect to individual such that the same individuals are included in both training and testing. Perhaps I'm misinterpreting? Please clarify here and in the Methods the details and interpretation of these cross-validation results.

153. Evaluating responses to energy development in absence of other covariates that are relevant to selection and might also be collinear with the placement of energy development sites could lead to confounding or spurious results in univariate models. I understand the reasons for doing so here but some attention to this concern may be warranted.

158. How were univariate responses to energy development incorporated into predictions from multivariable models without those covariates? I didn't see this step explained.

160. Please clarify the metric of consistency, presumably variation across years?

162. Comparing the consistency of 'premium' summer habitat to 'suitable' winter habitat seems to be a bit misleading given the different cut-off percentiles would seem to enforce such a result by definition. If the goal is to compare variation in consistency I would suggest using the same (or both) cut-off rules for each season.

167. Is it a bit circular or redundant to quantify habitat using time-varying and mechanistic covariates such as NDVI, IRG, RAP biomass products, etc.... which themselves are intended to capture the meaningful links between abiotic conditions (drought, temperature) and deer fitness (forage quality) and then include predictions of that model in another model that also includes weather metrics of temperature? The authors do clarify that temperature and winter severity can have effects on deer not mediated by forage quality, but it isn't fully fleshed out in the test of drivers of age ratios how weather conditions are included both in the habitat predictions and in the simple temperature, SWE, severity metrics. The results of this model seem to indicate that weather is more important than habitat, but perhaps such a result is even a bit too understated given weather conditions are included in the habitat predictions. If such distinctions can't be quantified with this approach, it might at least be worth further discussion.

197. I might also suggest some caution when interpreting selection as indicative of fitness benefits for a species with such rigid behavioral fidelity.

206. I'm not sure that this cited paper (Monteith et al. 2014) evaluated the direct effects of heat stress over indirect effects or temperature mediated through other mechanisms (e.g., forage) as described here.

236. The annual variation in predicted habitat (49-73% of landscape) seems like an interesting and useful result itself. I would suggest highlighting this aspect of the study as a result to be featured and can imagine some interesting figures along these lines!

308. A primary concern with these methods is the use of 3rd order selection models to predict statewide habitat patches. These models predict the relative probability of a site being used relative to available habitat within deer home ranges (i.e., given the site is within a home range). When mapping such a model across the entire state, one is essentially assuming every pixel is already in suitable habitat enough to be within deer home ranges (during both summer and winter) and predicting local use within that home range. By sampling availability only within deer home ranges, the data likely exclude some combinations of environmental conditions that occur outside of deer home ranges. Other approaches such as second-order habitat selection (selection of home ranges within broader study area landscapes) or combinations of 2nd and 3rd order selection would seem more appropriate for predictions at this broad scale. Conceptually this seems problematic but practically it is possible that the models still function reasonably well if home ranges were large enough to capture a wide

variation of conditions. In terms of practical risks of going forward with the current analysis: 1) the model may make predictions beyond the environmental conditions that were sampled (which can be avoided in various ways), and 2) the model may overestimate or incorrectly characterize fine-scale habitat in areas of the statewide extent that are not otherwise occupied by mule deer home ranges within each season, which should at minimum be discussed.

410. The use of just 2 somewhat arbitrary cut-offs for delineating binary habitat maps is reasonable but seems to potentially over-simplify or reduce the full amount of information on the table here. The binary depiction of habitat is obviously an over-simplification and the Table S3 shows that the results of age ratio models are fairly sensitive to the choice of cut-offs used to categorize. It just leaves me a bit unsatisfied and I'd appreciate discussion and/or exploratory analysis characterizing how these important decisions affected results.

416. Another very important methodological detail that I was missing in the Introduction, Results, and Discussion is that these age ratio data were measured in December. Thus, these results are capturing winter effects on maternal condition but not winter effects on over-winter fawn survival, which has otherwise been emphasized as a key demographic rate driving mule deer population dynamics in the literature. Throughout the manuscript (Abstract, Intro, Results, Discussion), more attention should be given to clarifying this detail... that any mention of 'recruitment' is recruitment only through the summer/fall and not through the entire first year of juvenile survival that includes over-winter survival. As is I find the current writing to be misleading along these lines.

Version 1:

Reviewer comments:

Reviewer #2

(Remarks to the Author)

I felt that the authors did a wonderful and thorough job revising the manuscript to address my and other reviewer comments. In particular, how they reframed their results/discussion surrounding variation in habitat suitability across seasons. I thought the addition of habitat suitability trends across mule deer ranges was a nice extension that could prove useful for many researchers/managers in the region. The authors also added some much needed context throughout that greatly improved the clarity of the manuscript. I have no further comments. Well done and I look forward to seeing this impressive work published!

Reviewer #3

(Remarks to the Author)

No further comments. Thanks to the authors for their time addressing comments and congratulations on a nice paper!

Open Access This Peer Review File is licensed under a Creative Commons Attribution 4.0 International License, which permits use, sharing, adaptation, distribution and reproduction in any medium or format, as long as you give appropriate credit to the original author(s) and the source, provide a link to the Creative Commons license, and indicate if changes were

made.

Reviewers' comments:

Reviewer #1 (Remarks to the Author):

This was a well written and innovative paper that was interesting and enjoyable to read. The author's approach is innovative, and the strength of this paper lies in the large spatial scale, the ability to account for variation over time at that scale, and the link between habitat and abiotic factors to the age ratios/recruitment for mule deer. In my opinion this will contribute to the future of ungulate research in general, as well as mule deer research and management in particular. The analysis was robust and well thought out, and while I have one comment on the analysis, most of my comments are about framing and interpretation and are relatively minor. I look forward to seeing this published.

We are grateful for the reviewer's feedback and have provided responses to each item below.

Major comment:

The only concern I have about the analysis is the correlation between the winter and summer abiotic variables used in the age-ratio model and the variables used in the RSF to generate the suitable and premium habitat layers. Summer precipitation and temperature are likely to be correlated with the NDVI and the peak green-up metrics, and the authors even found that summer precipitation was correlated with the entire habitat layer. I mention this because higher summer temperatures may have an additive effect of both shrinking the amount of premium habitat and causing heat stress and other heat related issues, as mentioned by the authors on lines 208-211 in the context of extreme drought. I think the authors need to mention this possible correlation in the methods as well as the discussion in the context that the effect of temperature may be even larger than found by the study. I would also recommend that the authors show that temperature is not correlated with the NDVI metrics used in the RSF analysis.

We appreciate your comment. We agree that abiotic conditions can influence habitat quantity and quality and thereby influence mule deer survival indirectly. We could not directly calculate correlation between individual variables informing the RSF and the abiotic variables used in the age ratio models because of differences in data resolution. The RSF is fit with fine scale resolution data whereas factors influencing age ratios like temperature are summarized at the herd unit level – the level at which age ratio counts take place.

**For the age ratio model, we found a minor negative correlation between premium**
**summer habitat and summer temperature ($r = -0.20$), however not a correlation**
**that would lead to exclusion from the age ratio model as was the case for the**
**precipitation variable (mentioned by the reviewer, $r = 0.58$). As we know**
**temperature plays a role in these forage metrics, we added a sentence in the**
**discussion to better indicate this potential influence. Precipitation, given its**
**stronger correlation with summer habitat, likely has a more robust indirect**
**influence on the forage metrics in the suitability model than temperature. We**
**added a paragraph in the discussion that addresses precipitation more directly.**

Minor Comments:

Lines 63-65: This sentence is unclear to me. Are you saying that there is strong concern
that abiotic factors will degrade habitat for ungulates? If so state that more clearly. If you
are stating that habitat quality (and by that I assume you mean forage quality) is
strongly linked to ungulate dynamics and so any declines in forage/habitat quality are
concerning for the management of abundant ungulate populations I would state that
more clearly.

**We edited this paragraph, deleting the sentence that caused confusion and**
**rephrasing the statement to better fit the ending of the paragraph. Edit: “*Given***
***the strong influence of abiotic conditions and habitat quality on demography,***
***degradation in conditions poses significant challenges to supporting populations***
***of many ungulate species.”***

Line 73 – I would say management concern because there is little chance of actual
extirpation of mule deer populations, but opinions may differ

**We prefer using the term ‘conservation’ here as the cited reference (Western**
**Association of Fish and Wildlife Agencies) also uses this term explicitly.**

Lines 104-105 – This sentence highlights the issue I raised in the comment above, that
winter severity and summer temperature/precipitation are inextricably linked to the
quality of the habitat. This requires a bit more acknowledgement by the authors that
these two are linked and the abiotic may influence that amount of available habitat, but I
still think the analysis approach is robust

**Please refer to our response in the ‘Major Comment’ section of this review. We**
**address this topic in detail there.**

Line 129 - There were some exceptions to the concept of “different by region but similar

in direction” in the sagebrush (trend for positive in Northeast) and heat load index (trend
for negative in Northeast) variables that were not addressed in the result text

**The 95% credible intervals for sagebrush and heat load index overlapped zero.**
**The median coefficient value may be positive or negative for each respectively**
**however statistically we found no detectable effect given the credible interval**
**width. We updated the language to indicate the concept of “different by region**
**but similar in direction” applies to “statistically detectable effects”.**

Lines 146-149 – This is definitely a pretty moderate correlation for this validation,
probably because of the large variation between the regions, but I would mention this
variation more than claim that this moderate correlation is indicative of consistent results
and instead say that even with the noise in the models from regional variation it was still
an informative covariate for the age-ratio models

**We revised the k-fold cross validation of the RSF models following the methods**
**of Boyce et al. (2002). This method provides more intuitive information regarding**
**predictive performance of the RSF models. Higher correlation indicates better**
**performance. Spearman rank correlation results indicate that the models**
**predicted cross-validated use locations well (summer: $r = 0.91$, $P < 0.01$; winter: r**
**$= 0.85$, $P = 0.03$).**

Line 159-160 – very interesting result and one that informs the debate on the
importance of summer vs winter habitat

**Thanks, we agree!**

Lines 174-178 – I would lead with lines 177-178 and then discuss the magnitude of
summer temp to make it easier to follow the findings and see the overall magnitude and
importance of summer temp

**We reordered this section as suggested.**

Lines 205-206 – Given the importance of summer temperature in the model results I
recommend adding another sentence and expanding possible heat stress impacts on
recruitment

**We added additional text as requested: “*Mule deer pregnancy rates, though***
***typically high (25-year mean = 98%, Monteith et al. 2014), depend on the doe's***
***energy intake before and during breeding (Tollefson et al. 2010), and***
***temperature-related impacts resulting in negative energy balances entering the***

***breeding season could potentially lower pregnancy rates. Temperature plays a***
***role in forage quantity and quality patterns (Wood et al. 2021). Thus, the full***
***accounting of the effect of temperature may be larger than estimated given the***
***inclusion of multiple forage covariates in the habitat suitability analysis.”***

Lines 220-225 – If there is any information on how long deer herds take to recover from
a winter event of this magnitude it would good to include here. I think some of the focus
on winter range vs summer range is because severe winters have a major and visible
impact on mule deer herds while poor summer habitat leads to more gradual declines

**We added an additional sentence to provide the context of recovery for the**
**example referenced in this comment. Edit: “*Age ratio surveys during the winter of***
***2024-2025 suggest reproductive recovery of this population after two years of***
***mild winter conditions and higher summer fawn survival (WGFD 2025). Despite***
***signs of increases in reproductive success, mule deer abundance across***
***Wyoming remains 50% below the 40-year average.”***

Informing management of mule deer – It may be beneficial to more explicitly
recommend that if agencies have limited budget for habitat restoration or management
to focus on summer range. Since winter range was consistent across years monitoring
winter habitat while restoring summer habitat seems like it would be better to slow mule
deer decline. The results of this paper also indicate, to me at least, that summer habitat
restoration will only slow the decline and it would be hard pressed to reverse the decline
if summer temperature is the strongest ecological driver of recruitment.

**We added emphasis on this to the abstract: “*Targeted management supporting***
***forage quantity and quality, especially on summer range.”*** At the end of the
**discussion, we also place emphasis on the importance of summer habitat**
**restoration: “*Potential high value priorities include continuing to build on efforts***
***to improve habitat quality in summer, with particular emphasis on temperature***
***refugia and forage robust to drought.”*** We also indicate the need for preserving
**winter habitat: “*Preserving the current relatively high integrity of habitat in the***
***identified winter use areas will also support higher fawn:doe ratios.”***

Line 301 – A word seems to be missing here “The _____ of these study herds...”
**Thanks for the catch. This sentence now reads: “*The home ranges of these study***
***herds are defined by diverse landscape characteristics including elevational***
***gradients, variable vegetation and faunal assemblages, land ownership patterns,***
***and energy development”***

Lines 357-358 – Add slightly more information here so this is repeatable. Did the effect
decay to 0 at 500m? Was this linear decay or another type of decay function?

**We have added text to clarify methods: “...based on negative exponential decay
functions where the effect declines to zero at 500 m...”**

Lines 373-374 – Table S2 referenced in the text is not a table of models but is actually
the selection coefficients for covariates, as correctly referenced on line 130. The table
with the a priori models needs to be added to the Appendix

**Thanks for catching this sentence. This statement is an artifact of a prior draft
and should be deleted as it was intended to reference the methods for modeling
age ratios not RSFs (age ratio modeling methods now appear later in the methods
section).**

Nice work on the figures. Clean, clear, and informative

**Thanks!**

Reviewer #2 (Remarks to the Author):

Overall, I commend the authors on what will no doubt be an important contribution to
the field of ecology as well as the management and conservation of ungulate
populations. I found the study’s stepwise progression, from assessing mule deer
resource selection, to using those selection results to measure habitat suitability, to
finally linking habitat suitability and other environmental variables to fawn:doe ratios, to
be both a creative and intuitive approach. The fact that this study was also done with
such an extensive dataset, spanning almost 1500 mule deer over 22 years from multiple
herds across Wyoming, makes it even more impressive. Furthermore, the analytical
approaches used to conduct this study were, in my eyes, statistically sound and
defensible. This section of the manuscript was also very strong and well written, with a
clear and comprehensive overview of the methods. Finally, the tables and figures
throughout the manuscript and SI were informative and aesthetically strong. It was clear
that significant thought, time, and effort went into producing them.

With that being said, I was a bit disappointed with the quality of the writing throughout
parts of the main text, especially throughout the Introduction and sections of the
Discussion. Currently, the main text is rife with run-on sentences that attempt to string
together too many ideas. These run-ons, combined with cases of poor sentence
structure and transitions, made it difficult to follow the manuscript at times. Without

considerable revisions to the writing, the main text does a disservice to all the fantastic
work the authors put in to amassing such an impressive dataset and conducting what I
believe to be an interesting and broadly impactful study.

As a result, I offer many line-by-line comments below, most of which apply to textual
edits, in the hopes that they improve the quality of writing in the manuscript.

**We appreciate the reviewer’s compliments and feedback and have responded to**
**each point below. Thanks for your constructive approach to this review.**

Abstract

L42-44: Consider changing sentence to following: “Identifying drivers of mule deer
demography could inform habitat restoration. However, linking habitat quality with
variation in vital rates is challenging and often done indirectly through proxy metrics.”

**We edited this sentence to align with the reviewer’s suggestions: “*Identifying***
***drivers of mule deer demography could inform habitat restoration. However,***
***linking habitat quality with variation in vital rates is challenging and often done***
***indirectly using proxy metrics.”***

L45: Replace “We apply a novel approach combining...” with “We combine...” to avoid
using qualifiers like new and novel as the journal suggests. Also cuts down on words to
get you closer to the 200-word limit (currently at 209) if necessary.

**Addressed as suggested.**

L46: Minor, but I would consider just using fawn:doe ratios throughout instead of age
ratios. Then readers don’t have to make the extra jump each time to remember age
ratios is only referring to fawn:doe ratios, as opposed to all variation in age within
populations.

**We appreciate the suggestion. Age ratios are commonly used synonymously with**
**fawn:doe ratios in deer (calf:cow in elk etc.) and for this reason we prefer to keep**
**the phrasing as is. However, we added additional reminders in each section of the**
**manuscript indicating age ratio refers to fawn:doe.**

L46: replace “data of” with “data from”

**Done.**

L48: replace “then modeled” with “model” to maintain tense of rest of sentence.

**Done.**

L51-57: Nice closing to your abstract that highlights the impact of your work, as well as
it’s only growing importance in future decades! Consider some minor suggestions to
improve the flow of the last sentence: “Our findings also show mule deer avoid areas
with high densities of oil and gas extraction. By delineating important mule deer habitat,
we offer spatial tools for development siting and mitigation across the western United
States.”

**This is a great suggestion. Incorporated. Thanks.**

Introduction

Overall, I found the introduction difficult to get through. It contains valuable information
pertinent to the research objective, but most of it is overwhelmed by run-on sentences
that attempt to convey too many points at once. As a result, the introduction lacks a
cohesive flow and fails to draw the reader in. Most of the introduction would benefit from
a focused effort to split longer sentences into multiple, and streamline language to
improve brevity while also taking the time to provide clarity and context for the reader. I
offer some examples below of how this could be done:

**Thanks.**

L63-65: This sentence is quite vague and feels repetitive. Not really understanding what
your message is. Maybe this better conveys what you’re trying to say?: “Abiotic factors
are strongly connected to the biotic world. Changing climates and landscapes lead to
habitat alteration, which in turn effects ungulate populations that depend on said habitat.
These connections motivate current concerns over the conservation of ungulate
populations amidst changing habitat.”

**We edited this paragraph, deleting the sentence that caused confusion and**
**rephrasing the statement to better fit the ending of the paragraph. Edit: “*Given***
***the strong influence of abiotic conditions and habitat quality on demography,***
***degradation in conditions poses significant challenges to effective management***
***of many ungulate species.”***

L65-70: Wonderful summation of the major ways in which changing habitats are
affecting ungulate populations.

**Thanks.**

L71-74: Since you have plenty of words to work with, you might consider splitting this
into multiple sentences. Because the paper focuses on mule deer and jumps into them
fast, I think readers would appreciate a couple of sentences going into more depth on
how they are economically and ecologically important. Highlight some of what's
discussed in the two papers you cite here. Then you can transition into their decline with
something like "However, mule deer are experiencing declining numbers across their
range due to..." And assuming the reason is habitat change, you could then add how
important it is to accurately quantify what habitat means for mule deer and end the
paragraph with that. This would segue into the following section nicely where you
discuss attempts to identify characteristics of mule deer habitat.

**We edited this sentence to incorporate more details as suggested by the reviewer.**
**Edit: "*Declines in mule deer populations can have both ecological and economic***
***consequences. For example, reduced browsing pressure may alter vegetation***
***communities, leading to shifts in plant composition and structure (Singer and***
***Renkin 1995). At the same time, these declines can affect the economy, including***
***the millions of dollars spent annually on big game hunting (Bergman et al. 2011)."***
**We also edited the rest of the paragraph for logical flow.**

L74-79: Move comma after "White et al 2025)" to after "across scales and seasons"

**Done.**

L82-83: Opening to the sentence feels clunky, consider "While this complexity makes it
hard to define "suitable habitat" for mule deer, seasonal..."

**Good suggestion, thanks. Addressed as suggested.**

L89-92: This sentence feels fragmented and disjointed to where I'm not entirely sure
what you're trying to say. What is meant by "habitat selection indicative of security by
the mother during early life" and how does this relate to the rest of the sentence and the
point made in the prior sentence? I suggest rewriting this sentence to flow more
logically, by first mentioning how habitat selection has been shown to influence body
condition, and then how body condition has been connected to variation in vital rates.

**We edited the sentence to clarify the intent. Edit: "...*habitat selection for greater***
***concealment by the mother during early life...*". We also reorganized this**
**paragraph to help with the flow.**

L100-114: This paragraph does a nice job succinctly summing up your study. Well done!
However, because your methods come at the end of the paper, I think you need to
move some key details either to this paragraph or within the “Results” section (this
might be a more natural fit). For instance, I wasn’t aware that 1) fawn:doe ratios were
collected at the start of winter in December, and 2) when you say winter habitat and
temperature, you’re referring to winter t-1 during doe gestation, rather than the first
winter experienced by the fawn. It should be easy to weave this in earlier than methods.

**Thanks. We understand the reviewer’s recommendation for a bit more details at**
**the end of the introduction to give context to readers. We made an edit to indicate**
**age ratios in this study represent: “...an index of recruitment of fawns into the fall**
**population.” We also added an additional statement to be more specific about the**
**lifecycle periods coinciding with the summer and winter conditions in our study:**
**“Given that age ratio surveys are primarily conducted in late fall we sought to**
**assess the relative influence of post-birth summer conditions experienced by**
**fawns compared to the winter conditions fawns were exposed to during**
**gestation.”**

**Results**

Great results section overall. Succinct and clearly describes the key findings of the
study. Just a few minor suggestions below.

**Thanks.**

L117: Since this follows the “Methods at the end” structure, make sure to define any
acronyms mentioned for the first time in your Results/Discussion rather than in your
Methods. For example, define GPS here instead of in Methods.

**Thanks. Done.**

L127: Again, because Methods come at the end, I would open this and subsequent
sections with a quick intro sentence stating what was done. Just to prime the reader for
the results they’re about to look at. Here, I would open with a quick list of the covariates
you considered. One option: “We examined mule deer resource selection of various
landscape covariates deemed a priori to be important. In summer, this included... In
winter, this included...”

**We understand the suggestion by the reviewer and have added text at the**
**beginning of the RSF results paragraph to front load readers with a little more**
**detail. Edit: “We analyzed mule deer habitat selection in relation to a suite of**

***landscape covariates identified a priori to broadly represent factors such as***
***forage availability, thermoregulation, and predator or human avoidance.***”

L133: Consider adding a few citations after “aligns with the literature” to support your
claim.

**We edited this sentence as suggested.**

L145-149: Consider restructuring this long sentence and breaking it into two. Something
like this “By using the conservative approach of randomly subsetting data into folds
without respect to individual identity and used-available status, cross-validation scores
indicated moderate correlation among folds (0.51-0.56). Thus, demonstrating that our
models have enough statistical power to overcome such limitations and produce
consistent results.”

**Agreed, addressed as suggested.**

L150: add “showed” or a similar word in between “GPS data” and “that most”. You can
also remove the hyphen from GPS data.

**Done. Added ‘indicated’.**

L159-160: Since you already say “higher variability” at the start of the sentence, you can
keep it simple and just say “in summer than in winter,” as it’s already implied winter was
less variable.

**Done. Edited as suggested.**

L161: replace “is” with “was” to be consistent with tense.

**Done.**

L159-165: While your use of premium for summer and suitable for winter makes sense
to me for the purpose of your analyses, is it fair to also apply this to your discussions
about seasonal variability in suitable habitat? In other words, is it defensible to compare
differences in variability when you’re using a different quality threshold across seasons?
For example, I would think the higher quality “premium habitat” would inherently be
more variable than the lower quality “suitable habitat,” especially if meeting this
“suitable” designation includes higher quality habitat designations like premium, since
that would also be considered suitable. On that point, was any habitat classified as
“premium” in winter? If so, it might be interesting to see that variability compared with
summer or simply compare suitable with suitable. I also acknowledge that this is a very
minor concern, as I assume the “suitable” winter designation is roughly equivalent to the

“premium” summer designation, since habitat suitability is expected to be much lower
overall in winter.

**We agree with the reviewer’s perspective, and we reframed this results**
**paragraph. We put more emphasis on the spatial variation in habitat suitability**
**(not classified suitability i.e., suitable/premium). We also added information on**
**the trends in habitat suitability at different spatial scales. Edit: “We observed**
***similar ranges in predicted habitat suitability across seasons (Figure 3a-b);***
***however, suitability values were twice as variable across space during summer (σ***
***= 0.14) than winter ($\sigma = 0.07$). Between 2000 and 2023, across the study area***
***habitat has an average classification of suitable for 22.35 ($\sigma = 4.51$) years in***
***winter use areas and as premium for 15.11 ($\sigma = 8.99$) years during summer (Figure***
***3 C and D). When comparing trends across regions used in the RSF analysis we***
***found a negative trend in the amount of premium summer habitat in the***
***northeastern region ($\beta = -0.03$, $p < 0.001$) and no trend in the other regions.***
***Summarized further, five herd units located in the northeastern portion of***
***Wyoming exhibited declines resulting in an observed loss in premium summer***
***habitat area of 4179 km² (1613 mi²) from 2000 to 2023 (Figure 4, Appendix S1:***
***Table S6). We found mixed results across herd units regarding trends in suitable***
***winter habitat with three herd units showing positive trends and three showing***
***negative trends over time(Appendix S1: Table S6). The northeastern region***
***exhibited reductions in suitable winter habitat ($\beta = -0.03$, $p < 0.001$) with declines***
***particularly around the North Converse herd unit ($\beta = -0.05$, $p < 0.001$). The***
***modeled trends of winter habitat are likely mainly driven by the severe 2022–2023***
***winter, as we found a net increase of 442 km² in suitable winter habitat in 2023***
***compared to 2000. The proportion of area classified as suitable in winter***
***remained ≥ 0.95 annually across the northeastern region. We found no regional-***
***level trends in winter habitat suitability in other portions of Wyoming.”***

L167-169: Another place where you might consider clarifying that winter habitat and
temperature

**This comment appears cutoff, but we did our best to address according to the**
**intent.**

L171: Briefly define “looiic” here since this is the first mention

**This sentence has been edited to: “The model with all variables ranked lowest in**
**the leave-one-out cross-validation information criterium (looiic, Appendix S1:**
**Tables S3-S4), and we made inference from these results (Figure 4).”**

**Discussion**

Aside from some suggestions to improve the clarity of the writing, I think you have a
nice discussion that hits most of the key points of the study and how it extends to the
management/conservation of mule deer. With that being said, since you didn't factor in
predation risk or overwinter survival and true recruitment in your analyses, it would be
beneficial to devote a paragraph to these topics/limitations of your study (i.e.,
mentioning the importance of predation/predation risk for shaping ungulate movements
and survival, especially for juveniles; any thoughts on how future work could be
improved by trying to incorporate predation/risk; importance of measuring overwinter
survival and true recruitment for understanding impacts of habitat and resource
selection on population demographics, what do your significant results about winter t-1
say about how strong of an effect the current winter may have on recruitment/survival?).

**We have added text to the discussion about both predation and direct**
**measurements of vital rates. For predation, we added: “Detailed information on**
**predation risk was not available at the scale of our study and is absent from the**
**resource selection analyses. Nonetheless, some covariates used to model habitat**
**suitability—such as topography and vegetation cover—may serve as proxies for**
**spatial variation in predation risk. Future research could benefit from explicitly**
**incorporating predator density or predator-specific resource selection patterns .”**

**The timing of the age ratio surveys and the nature of the metric—reflecting**
**recruitment into the fall population—limit our assessments to covariates from**
**winter t–1. Although it is reasonable to expect that a severe current winter would**
**result in lower age ratios if age-ratios were recorded later in the winter, we could**
**not directly assess this. We agree that measuring overwinter survival and true**
**recruitment can be useful for evaluating especially fine scale effects of habitat**
**and resource selection on population demographics. We believe our approach**
**illustrates that age ratios can be informative about these relationships across**
**broad extents.**

L184: Remove novel approach to align with journal guidelines. Could instead say “By
combining habitat selection with climate-related effects, we identify...” In a similar vein,
while I recognize that it is important to showcase how your study has moved the bar
forward (and I agree that it has done a nice job of this), I don't think this needs to be
pushed as much as it currently is in the manuscript. For example, at several points you
discuss that previous studies have only ever made indirect links between habitat
quality/selection and vital rates/population dynamics or have only used proxy metrics. I
would generally refrain from making these types of claims given the vastness of the

literature. Undoubtedly, this link has been made somewhere on some scale. More so,
even though your connection of habitat selection to fawn:doe ratios is getting closer to
linking habitat to true demography/vital rates, fawn:doe ratios are still just a proxy for a
vital rate, especially since you aren't capturing overwinter survival or accounting for
pregnancy rates within herds. In my opinion, I would just let your amazing data speak
for itself and focus on all the relevant findings.

**Thanks for the suggestions. We agree softening the language showcases the**
**results of this study just as well. We have worked to address this comment**
**broadly throughout the manuscript. Related to this section at the beginning of the**
**discussion, we rewrote the first few sentences to simply contrast our work from**
**prior research.**

L190-192: This sentence is also a bit clunky. Consider this revision to improve flow:
"However, preserving or enhancing habitat—such as by managing invasive annual
grasses—could help buffer against the negative effects of rising summer temperatures."

**Thanks, edited as suggested.**

L193-195: The transition here is pretty poor and, as a result, feels random and out of
place. Maybe this works better? "Another growing concern is that mule deer have high
fidelity to seasonal ranges and migration routes relative to other ungulates (Morrison et
al. 2021). While this aligns with the critical assumption that resource selection patterns
from RSF analyses are static, it also means mule deer are more susceptible to climatic
and anthropogenic changes."

**Thanks, edited as suggested for flow. See next comment also.**

L197-200: This sentence is redundant to your earlier sentence describing high fidelity to
seasonal ranges and migration routes. I would recommend removing it and ending on
your point that this makes mule deer more susceptible to change.

**Importantly, this sentence brings in the fidelity to migration behavior (migrant vs**
**resident) in addition to migration route fidelity. We integrated the last few**
**sentences to reduce redundant language: "*Mule deer have high fidelity to***
***seasonal ranges, migration routes, and migration behaviors (migrant versus***
***resident) relative to other ungulates (Sawyer et al. 2019, Morrison et al. 2021).***
***This aligns with the critical assumption that resource selection patterns are***
***static, but it also suggests mule deer may be more reliant on seasonal habitat***
***conditions than other ungulates and consequently more susceptible to climatic***
***and anthropogenic changes.*"**

L201-225: While this is a great section that adds a lot of relevant information on how
changing weather and climate is expected to impact mule deer recruitment, I think it
would benefit from placing all this in the context of your results. For example, if summer
temperature and drought are expected to increase in coming years which should
negatively impact mule deer based on these studies. How do your findings in this study
directly support these claims? Always try and relate back to your own study throughout
each discussion section.

**We added the following statement to speak more directly to the broader message**
**of this paragraph. Edit: “*Temperature negatively impacted mule deer age ratios at***
***a rate 1.3 to 2.5 times greater than the positive influence of habitat suitability.*”**
**This emphasizes the effects of temperature outweigh the effects of habitat**
**suitability on age ratios and sets up for the implications.**

L211-214: Another run-on sentence that’s difficult to follow. Consider this revision:
“Given these adverse drought effects that place pressure on higher elevations to
provide refugia from summer heat, adopting management practices that support soil
moisture retention and drought-resistant perennials may be especially beneficial for
mule deer.”

**We edited this statement to align with the reviewers comment: “*Given the adverse***
***drought effects, adopting management practices that support soil moisture***
***retention and drought-resistant perennials may be especially beneficial for mule***
***deer.*”**

L218: Add a comma after “atmospheric flows” and just say “which is a hypothesized
mechanism for...”

**Done.**

L230-233: Remove “is a novel approach and...” Also, while it’s certainly affecting
recruitment, you might caution against using the term when applying to your study,
since you only really capture an index of fawn survival up to but not including winter
(which can be pretty low depending on winter severity), and thus, aren’t measuring true
recruitment into the population.

**Done, removed “is a novel approach” and softened language to say “aspects of**
**recruitment”.**

L261-263: I would edit “We found avoidance of areas with increased amounts of energy

infrastructure by mule deer and past studies have also shown an avoidance of roads
particularly during the installation of energy development when worker activity is highest
(e.g., Sawyer et al 2006).” in the following way: “We found that mule deer avoided areas
with more energy infrastructure, aligning with past studies that showed an avoidance of
roads, particularly during the installation of energy infrastructure when worker activity
was highest (e.g., Sawyer et al 2006).”

**Thanks, edited per suggestion.**

L266-269: commas after “annually” and “traffic”

**Done.**

L271-273: hyphenate “third order” and “fine scale”

**Done.**

L281-282: simply saying “support demographics” is vague. I would replace with
something like “support higher fawn:doe ratios.”

**Edited as suggested for specificity.**

L284-286: You might replace “oil and gas” with “infrastructure for oil and gas extraction”
just to be clear what is meant.

**Edited to include the term “infrastructure”.**

L287: comma after “space and time”

**Done.**

In the last paragraph, you might consider adding a few sentences about the broader
implications of your work for ungulate conservation. In other words, do your findings
about mule deer across Wyoming apply more broadly to their entire range across
western NA, can they shed light on how we might expect other ungulate species to
respond with increasing global change? Just some thoughts for highlighting the broader
impact of this great study!

**We appreciate the reviewer’s encouragement. We’ve broadened the results and**
**discussions to highlight more of the temperature and precipitation effects. We**
**believe this provides readers with more information about future conditions.**

Methods
Fantastic methods section. With great clarity and organization, you provided a very
comprehensive overview of your analytical approach. Well done! Below are a few minor
text edits and suggestions.

**Thanks!**
L301: remove “of these”

**The sentence was missing context and now begins: “The home ranges of these**
**study herds...”**
L303-304: hyphenate “confer dominated”

**Done.**
L314: since you only use this once in the manuscript, I would replace “NSD” with “Net
Squared Displacement” to save unfamiliar readers the trouble of figuring out what this
means.

**Done.**
L324: typo, replace “availably” with “availability” or “available” depending on intention.

**Thanks for the catch. Now reads as “availability”.**
L336: replace “as proxies for forage” with “as proxies for the”

**Done.**
L360: hyphenate “90 m”

**Kept as 90 m to align with rest of text.**
L409-411: I might just be reading this incorrectly, but if these thresholds are below \bar{x} ,
wouldn't it be more accurate to describe these as the 40th and 10th percentiles for
defining premium and suitable habitat, respectively? Also, it might be worth moving
these details to the results section so readers know how you are defining suitable and
premium habitat, since this decision forms the backbone of the study.

**Thanks for the catch! The values as pointed out by the reviewer should be the**
**40th (containing 60% of the distribution) and the 10th (containing 90% of the**
**distribution). We corrected the text.**

L416-421: Here's where some of your details about the timing of summer and winter
covariates could be moved into the main text to orient readers.

**We added information about the timing of summer and winter covariate and their**
**biological relevance to the main text at the end of the introduction. Edit: “Given**
**that age ratio surveys are primarily conducted in late fall we sought to assess the**
**relative influence of post-birth summer conditions experienced by fawns**
**compared to the winter conditions fawns were exposed to during gestation.”**

L434-438: Interesting (and sobering) little piece of information here that you gleaned
from looking into this covariate that was ultimately dropped!

**Thanks.**

L463: replace “can be less” with “can be a less”

**Done.**

Tables and Figures

Figure 2: Consider adding summer and winter labels with figure letters as you did in
Figure S1

**Done.**

Supplementary Information

Great work on the SI. Figures and tables are very strong. Beautiful while providing all
the necessary information to support the main text.

**Thanks.**

Figure S2: If it's not too difficult, can you change the colors of the herds in the left plot to
match the colors in the right plot? Also, there seems to be a mismatch between the data
in the left plot and the right plot. The percentages on the right only add up to 93%,
whereas on the left they appear to add up to 100%. Also, your statement that 76%
summer in areas with higher proportions of suitable habitat suggests the Upper

Shoshone value should be 24% rather than 17%, yet this wouldn't match with the
current distribution on the left plot. Instead, it looks like Dubois should be more like 40%,
in which case the 76% should be 83%.

**Thanks for the catch! The correct value for Dubois is 40% with a total of 83% of**
**collared deer spending summer outside of the Upper Shoshone unit. This error**
**arose during manually labeling the figure.**

**Reviewer #3 (Remarks to the Author):**

The authors conducted resource selection analyses for a large dataset of mule deer
across Wyoming and then correlated predicted maps of seasonal habitat, along with
metrics of drought and winter severity, to December recruitment ratios. I found the
manuscript to be very well-written and the work to be thorough and carefully crafted. I
have a few questions geared at methods that might warrant further consideration prior
to publication as well as some minor editorial suggestions.

**Thank you for the insights. Our detailed responses to each point are provided**
**below.**

42-43. I found this 2nd sentence of the Abstract to be a bit too big a leap, requiring
some more detail or setup to jump to these complex themes such as demographic
drivers of decline, habitat improvement projects, and the challenges of linking metrics of
habitat quality to vital rates.

**We worked to streamline and generalize this section based on other reviewers'**
**comments as well.**

55. The phrase "oil and gas areas" is a bit vague.

**Done. Changed 'areas' to 'development'.**

136. The authors excluded migratory paths in their delineation of seasonal ranges,
which I think negates their direct relevance to surfing of green waves (which happens
during migration) despite migrating to areas with later peak productivity.

**Our inclusion of covariates that reflect green wave dynamics in summer habitat**
**models forms the basis of our original statement. Our intent was to point out that,**
**based on the RSF analyses, deer summer in areas that reach peak green-up later**
**(mentioned on the previous line to the statement in question). This supports**
**other findings of seasonal patterns in space use. Edited to clarify: "*This supports***
***findings that deer track the progression of forage growth and use seasonal***

***ranges in relation to peak resource availability.”***

146. I'm unclear how to interpret these cross-validation scores nor is enough detail
provided in the Methods to do so. Also, generally I find randomly sub-setting data by
individual to be more robust in delineating independent training and testing folds of data
than to create subsets without respect to individual such that the same individuals are
included in both training and testing. Perhaps I'm misinterpreting? Please clarify here
and in the Methods the details and interpretation of these cross-validation results.

**We revised the k-fold cross validation of the RSF models following the methods**
**of Boyce et al. (2002). This change brings our assessment of predictive**
**performance inline with the reviewer's suggestions and provides a more intuitive**
**result. Edit: “We partitioned data into five equal-sized folds, ensuring each fold is**
**a random subsample of individuals to maintain independence between training**
**and validation data. Predictive performance was assessed by ranking RSF scores**
**from the validation fold, binning them into habitat suitability levels, adjusting**
**frequencies by availability, and computing the Spearman rank correlation**
**between bin ranks and area-adjusted frequencies (Boyce et al. 2002). Spearman**
**rank correlation for frequency values by bins indicates that the models predicted**
**cross-validated use locations well (summer: $r = 0.91$, $P < 0.01$; winter: $r = 0.85$, $P =$**
**0.03).”**

153. Evaluating responses to energy development in absence of other covariates that
are relevant to selection and might also be collinear with the placement of energy
development sites could lead to confounding or spurious results in univariate models. I
understand the reasons for doing so here but some attention to this concern may be
warranted.

**We appreciate the reviewer's suggestion. To better share this information, we**
**have updated the text about modeling the effects of energy development to be**
**more detailed. We edited this text in the Methods for clarity: “Preliminary**
**assessment of GPS locations indicated many individuals did not encounter oil**
**and gas development within their home ranges (75% in winter and 68% in**
**summer). Such lack of availability of a given resource can induce bias in RSF**
**modeling. Therefore, we pooled individuals that encountered energy development**
**across regions under the assumption that response to this would be similar. We**
**fit univariate models to determine the effect of energy development on selection**
**for winter and summer periods and multivariate (all RSF covariates) models to**
**evaluate collinearity with other covariates. The estimated effect of energy differed**
**little between univariate and multivariate models therefore we used univariate**

***estimates in predictions. We fit univariate models to determine the effect of***
***energy development on selection for winter and summer periods and multivariate***
***(all RSF covariates) models to evaluate collinearity with other covariates. The***
***estimated effect of energy differed little between univariate and multivariate***
***models therefore we used univariate estimates in predictions.***

158. How were univariate responses to energy development incorporated into
predictions from multivariable models without those covariates? I didn't see this step
explained.

**We treated energy development as a direct additive effect and have updated the**
**text to reflect this clearly. See previous response to Line 153 about the calculation**
**methods.**

160. Please clarify the metric of consistency, presumably variation across years?
**Deleted 'consistent' – now simply states both coefficients mentioned were**
**negative.**

162. Comparing the consistency of 'premium' summer habitat to 'suitable' winter habitat
seems to be a bit misleading given the different cut-off percentiles would seem to
enforce such a result by definition. If the goal is to compare variation in consistency I
would suggest using the same (or both) cut-off rules for each season.

**We agree with the reviewer's perspective and reframed this results paragraph. We**
**put more emphasis on the spatial variation in habitat suitability in winter and**
**summer (not classified suitability i.e., suitable/premium). We also added**
**information about the trends in habitat suitability at different spatial scales to**
**provide more relevant details to the reader.**

167. Is it a bit circular or redundant to quantify habitat using time-varying and
mechanistic covariates such as NDVI, IRG, RAP biomass products, etc.... which
themselves are intended to capture the meaningful links between abiotic conditions
(drought, temperature) and deer fitness (forage quality) and then include predictions of
that model in another model that also includes weather metrics of temperature? The
authors do clarify that temperature and winter severity can have effects on deer not
mediated by forage quality, but it isn't fully fleshed out in the test of drivers of age ratios
how weather conditions are included both in the habitat predictions and in the simple
temperature, SWE, severity metrics. The results of this model seem to indicate that
weather is more important than habitat, but perhaps such a result is even a bit too
understated given weather conditions are included in the habitat predictions. If such

distinctions can't be quantified with this approach, it might at least be worth further
discussion.

**We added a paragraph in the discussion that speaks to the concept of indirect**
**effects; for example, precipitation altering forage thereby altering habitat**
**suitability and subsequently mule deer survival. Habitat suitability derived from**
**the RSF models does represent both the indirect effects of precipitation and the**
**direct effects of the availability of forage. Whereas in the age ratio models the**
**abiotic covariates represent more direct impacts on mule deer.**

197. I might also suggest some caution when interpreting selection as indicative of
fitness benefits for a species with such rigid behavioral fidelity.

**We have edited this section of text to now read: “Another source of concern is**
**that mule deer have high fidelity to seasonal ranges, migration routes, and**
**migration behaviors (migrant versus resident) relative to other ungulates (Sawyer**
**et al. 2019, Morrison et al. 2021). While this aligns with the critical assumption**
**that resource selection patterns are static, it also means mule deer may be more**
**reliant on seasonal habitat conditions than other ungulates and consequently**
**more susceptible to climatic and anthropogenic changes.”**

206. I'm not sure that this cited paper (Monteith et al. 2014) evaluated the direct effects
of heat stress over indirect effects or temperature mediated through other mechanisms
(e.g., forage) as described here.

**The reviewer is correct. Our understanding is Monteith et al. (2014) found**
**negative relationships between temperature and a variety of factors (body mass,**
**body condition, pregnancy etc) and hypothesized these effects manifested via**
**mechanisms like forage quality. We edited this statement to now read:**
**“Increasing summer temperatures can decrease the survival of neonates by**
**influencing their body condition and that of their mother while nursing (Monteith**
**et al. 2014).”**

236. The annual variation in predicted habitat (49-73% of landscape) seems like an
interesting and useful result itself. I would suggest highlighting this aspect of the study
as a result to be featured and can imagine some interesting figures along these lines!

**Thanks, we added a paragraph in the results about trends in predicted habitat to**
**flesh this out more clearly for readers.**

308. A primary concern with these methods is the use of 3rd order selection models to

predict statewide habitat patches. These models predict the relative probability of a site
being used relative to available habitat within deer home ranges (i.e., given the site is
within a home range). When mapping such a model across the entire state, one is
essentially assuming every pixel is already in suitable habitat enough to be within deer
home ranges (during both summer and winter) and predicting local use within that home
range. By sampling availability only within deer home ranges, the data likely exclude
some combinations of environmental conditions that occur outside of deer home
ranges. Other approaches such as second-order habitat selection (selection of home
ranges within broader study area landscapes) or combinations of 2nd and 3rd order
selection would seem more appropriate for predictions at this broad scale. Conceptually
this seems problematic but practically it is possible that the models still function
reasonably well if home ranges were large enough to capture a wide variation of
conditions. In terms of practical risks of going forward with the current analysis: 1) the
model may make predictions beyond the environmental conditions that were sampled
(which can be avoided in various ways), and 2) the model may overestimate or
incorrectly characterize fine-scale habitat in areas of the statewide extent that are not
otherwise occupied by mule deer home ranges within each season, which should at
minimum be discussed.

**All good points raised by the reviewer. Broadly speaking, these points are**
**precisely why we limited the geographic range of habitat in the winter RSFs**
**(cropped to predefined winter use areas). This is less of an issue for the summer**
**RSFs because mule deer are more spread out across the landscape experiencing**
**a broader range of conditions during this time as indicated by the Wyoming**
**Observation System of wildlife locations which shows widespread occurrence of**
**mule deer in summer. We completed a priori checks on the alignment of habitat**
**covariates in- and outside of individual home ranges and found broad overlap**
**indicating the large spatial extent of GPS-collar data used in this study captures**
**the range of habitat conditions statewide. Nonetheless, we still wanted to ensure**
**robust habitat predictions in areas where we had no mule deer data informing the**
**RSF models. This is why we implemented an approach that spatially weighted**
**those habitat predictions by proximity to the regions used in the RSF. Our**
**approach is based on the hypothesis that deer respond similarly to other deer**
**experiencing the same habitat conditions which is supported by the results of the**
**regional RSF models. We edited the portion of the Appendix where the habitat**
**prediction methods are discussed to include additional context of the decision**
**process and a *priori* checks mentioned here.**

410. The use of just 2 somewhat arbitrary cut-offs for delineating binary habitat maps is
reasonable but seems to potentially over-simplify or reduce the full amount of

information on the table here. The binary depiction of habitat is obviously an over-
simplification and the Table S3 shows that the results of age ratio models are fairly
sensitive to the choice of cut-offs used to categorize. It just leaves me a bit unsatisfied
and I'd appreciate discussion and/or exploratory analysis characterizing how these
important decisions affected results.

**We agree the cutoff for suitable versus premium habitat is a simplification but**
**one that indicates the importance of categorizing habitat at different levels to**
**inform demography (age ratios in our case). The hypothesis we were most**
**interested in testing was whether standard definitions of suitable was sufficient**
**versus refined assessments of preferred habitat. To that end we followed**
**previous work on these methods (Coates et al. cited in text) to guide our**
**implementation. However, readers (and importantly, managers) will have access**
**to continuous spatial predictions of habitat suitability via our data release from**
**which different cutoffs could be explored. We added discussion about trends**
**over time in uncategorized habitat suitability to explore this topic more.**

416. Another very important methodological detail that I was missing in the Introduction,
Results, and Discussion is that these age ratio data were measured in December. Thus,
these results are capturing winter effects on maternal condition but not winter effects on
over-winter fawn survival, which has otherwise been emphasized as a key demographic
rate driving mule deer population dynamics in the literature. Throughout the manuscript
(Abstract, Intro, Results, Discussion), more attention should be given to clarifying this
detail... that any mention of 'recruitment' is recruitment only through the summer/fall and
not through the entire first year of juvenile survival that includes over-winter survival. As
is I find the current writing to be misleading along these lines.

**We appreciate the comment and have no intention to be misleading about this**
**topic. We specify in the Methods that age ratio surveys occur in December, as**
**pointed out by the reviewer. Indeed, the age ratios in our study reflect fawn**
**recruitment into the late-fall population (used by Wyoming Game and Fish**
**Department managers to assess mule deer recruitment in Wyoming). We also**
**edited the text in the introduction to be more specific. Edit: "Our main goal was to**
**quantify the relationship between the amount of seasonal habitat available in the**
**study area and age ratios (i.e., fawn:doe ratios, an index of recruitment of fawns**
**into the fall population)..."**